# Proprioceptive integration in motor control

Erik Skjoldan Mortensen  and Mark Schram Christensen

*Department of Psychology, University of Copenhagen, Øster Farimagsgade 2A, Copenhagen, Denmark*

Handling Editors: Vaughan Macefield & Jacob Thorstensen

The peer review history is available in the Supporting Information section of this article (https://doi.org/10.1113/JP289835#support-information-section).

**Abstract figure legend** A virtual reality (VR)-based reaching task reveals how proprioceptive feedback affects the perceived and actual movement speed of active movements, as well as the perceived hand position. When an antagonist vibration was applied, subjects tended to overestimate their movement speed while also decreasing their actual movement speed. When visual feedback from the virtual arm was removed, a similar decrease in movement speed was observed, and subjects also undershot the target, indicating an induced offset in the perceived hand position. Similar effects in the opposite direction were observed with agonist vibration, although less consistently and smaller in size, whereas dual vibration produced intermediate bias effects, but no increased spread of endpoint errors. During control trials with no vibration, we observed effects similar to those induced by muscle vibration: faster movements were associated with underestimation of movement speed and overshooting the target, whereas slower movements were associated with the opposite.

**Abstract**  Muscle vibration alters both perceived limb position and velocity by increasing muscle spindle afferent firing rates. In particular the type Ia afferents are affected, which mainly encode muscle stretch velocity. Predictive frameworks of sensorimotor control, such as Active Inference

**Erik Skjoldan Mortensen** holds a master's degree in human physiology and is working on completing his PhD in cognitive neuroscience under the supervision of Mark Schram Christensen at the Section for Cognition and Neuropsychology, Department of Psychology, University of Copenhagen. As part of the Cognition, Intention and Action group, his research focuses on exploring proprioception, sensory inference and motor control.

This article was first published as a preprint. Skjoldan Mortensen E, Schram Christensen M. 2025. Proprioceptive integration in motor control. bioRxiv. https://doi.org/10.1101/2025.07.18.665517

and Optimal Feedback Control, suggest that velocity signals should inform position estimates. Such a function would predict that errors in perceived limb position and velocity should be correlated, but this prediction remains empirically underexplored. We hypothesised that an online evaluation of the integral of sensed velocity influences the perceived arm position during active movements. Using a virtual reality-based reaching task we investigated how vibration-biased proprioceptive feedback influences voluntary movement control and inference of arm position and movement. Our results suggest that muscle vibration biases perceived movement velocity, with downstream effects on perceived limb position and reflexive corrections of movement speed. We found that (i) antagonist vibration during active movement caused participants to overestimate their movement speed while also slowing down, (ii) movement speed and endpoint errors were correlated, with muscle vibration affecting both in congruent directions and (iii) adjustments in movement speed to muscle vibration are sufficiently fast to be reflexive. Together these findings support the hypothesis that proprioceptive velocity signals are integrated to augment inference of position, consistent with predictive frameworks of sensorimotor control.

(Received 31 July 2025; accepted after revision 18 February 2026; first published online 16 March 2026)

**Corresponding author**: E. S. Mortensen: Øster Farimagsgade 2A, Copenhagen, Denmark. Email: esm@psy.ku.dk

**Key points**

- During movement without visual feedback, the central nervous system (CNS) has access to both position- and velocity-based proprioceptive signals, which are used to estimate limb state.
- Muscle vibration biases the perception of limb position, as seen in the classically observed pattern of biased endpoint errors, through the stimulation of primary (type Ia) muscle spindles, primarily a velocity sensor.
- We investigated how proprioceptive velocity signals affect position estimation during movement by applying muscle vibration while measuring perceived movement speed, actual movement speed and endpoint errors in a virtual reality (VR)-based reaching task.
- We show that errors in perceived limb position and velocity are correlated during active movements, consistent with predictive frameworks of sensorimotor control.
- These findings support the idea that the CNS maintains a self-consistent estimate of limb state across both position and velocity domains.

## Introduction

To reach out and grasp an object we must infer our own body position and movement kinematics. If we plan for our hand to arrive at a specific end position with a particular velocity (e.g. zero), the action required to achieve this must account for the hand's position and velocity as it is moving towards the target. Central to this task is the continuous stream of sensory information processed by the central nervous system (CNS), with vision and proprioception being the most relevant for a reaching task. Although vision is not necessary (we can reach a target without looking at our hand) the sense of proprioception continuously supplies the CNS with information regarding the arm's movement parameters.

Proprioception describes the sensation of the position and movement of the body, which arises from a collection of peripheral mechanoreceptors found in the muscles,

skin and joints all around the body (Tuthill & Azim, 2018). Beyond giving rise to the perception of the state of the body these feedback signals are also tightly integrated with motor control, as observed in cases of peripheral deafferentation (Cole & Waterman, 1995). To limit the scope of our introduction to the subject we will here mainly consider the perceptual and behavioural effects of muscle spindle afferents, which are stretch-sensitive receptors found throughout the skeletal muscle system, where they innervate the non-contractile central region of the intrafusal muscle fibres (Proske & Gandevia, 2012).

This proprioceptive information is incorporated into motor control at several different hierarchies in the CNS, with corresponding types of possible movement corrections. These range from short-latency reflexes typically seen in the 20–45 ms range (in the upper limbs), through long-latency reflexes at 50–100 ms, with even longer response times being necessary for conscious and

deliberate control (Kurtzer, 2015). Although relatively simple spinal circuits mainly drive the former, the long-latency reflexes are trans-cortical and show evidence of being coordinated by an internal model of limb dynamics (Keyser et al., 2019; Kurtzer, 2015; Kurtzer et al., 2008; Nakazawa et al., 1997; Yamamoto & Ohtsuki, 1989).

Two of the primary peripheral sensory organs responsible for these reflexes are the type Ia and type II muscle spindle afferents (Proske & Gandevia, 2012). They carry rich information regarding limb movement, allowing decoding of imposed movement patterns (Albert et al., 2006; Roll et al., 2004). The type Ia afferent primarily signals the rate of change in muscle length, that is, a velocity signal (Albert et al., 2006; Roll et al., 2004), whereas the type II afferent primarily signals muscle length directly, that is, a positional signal (Roll & Vedel, 1982). In addition to being sensitive to lengthening of the receptor-bearing muscle both type Ia and type II afferents respond to vibration of the muscle/tendon by increasing their firing rate up to the vibration frequency or a sub-harmonic of it (Roll et al., 1989). This has prompted a large body of experimental research investigating the contents and inference of these proprioceptive signals. At a behavioural level it has been demonstrated that the vibration of a muscle can bias both the perceived limb position and limb velocity (Eklund, 1972; Gilhodes et al., 1986; Goodwin et al., 1972; Inglis et al., 1991; Roll & Vedel, 1982). It should further be noted that the Golgi tendon organ, in addition to both types of muscle afferents, is affected by vibration when the muscle is contracting (Roll et al., 1989), making it difficult to disentangle the perceptual effects originating from each receptor type independently. The type Ia does, however, respond particularly well, and it is generally believed that the primary perceptual effects of muscle vibration originate from this receptor type (Proske & Gandevia, 2012).

One of the primary techniques for investigating this phenomenon involves vibrating the muscles of a stationary limb and tasking the research participants with mirroring the perceived movement with the contralateral limb. Although researchers have explained their findings in such experiments in terms of either biased perception of limb position or velocity, it proves challenging to design experiments that allow for distinctive reports of perceived limb velocity, without also inherently reporting changes in limb position, and vice versa – and, in addition, the perceptual effects may depend upon which of these aspects the subject is tasked to focus on (Sittig et al., 1985). In all cases the induced bias, whether in dynamic joint movement or a change in a statically sensed position, is perceived to be in the direction that would have resulted from stretching the vibrated muscle, with the amplitude of the induced bias being positively correlated to vibration amplitude (Calvin-Figuière et al.,

1999; Roll & Vedel, 1982). Even complex illusions of limb movement have been demonstrated by 'playing back' microneurographically recorded Ia afferent firing patterns through vibration of muscles all around the ankle, allowing subjects to recognise specific movement patterns (Albert et al., 2006). These findings suggest that joint and limb movements are inferred from the combined signalling of afferents from all the muscles involved in controlling said joint.

Studies that have specifically investigated the effects of concurrent vibration of a flexor-extensor muscle pair in non-moving conditions have shown varied effects. Gilhodes et al. (1986) found that concurrent vibration tends to cancel each other out when the vibration frequency is kept identical, with effects appearing when the frequency of one motor is decreased, with the faster vibration motor then beginning to produce an effect, although it is smaller than if this vibration motor were used alone. Fuentes et al. (2012) and Chancel and Ehrsson (2023) have argued that concurrent vibration may instead degrade the available proprioceptive information, such that variable error increases, and less weight is placed on proprioceptive feedback during sensory integration, and, as an example, leading to increased belief in the rubber hand illusion.

When vibration is applied during active movements very similar perceptual effects are induced, as seen in passive conditions; when vibration is applied to the lengthening antagonist muscle, it will generally result in undershooting of a target, indicating that the change in joint angle is overestimated (Capaday & Cooke, 1981). Moreover when vibration is applied to a larger portion of the movement, the target is undershot to a larger degree, indicating a cumulative effect on the perceived position, while movement speed is proportionally reduced at the same time (Cody et al., 1990; Eschelmuller et al., 2023). In contrast applying vibration to the agonist muscle during active contraction has generally produced little or no measurable effect (Capaday & Cooke, 1981, 1983; Inglis & Frank, 1990; Macefield & Knellwolf, 2018). However a recent study reported that agonist vibration can lead to a small overshoot effect, although it remains weaker than the undershoot caused by antagonist vibration in the same task (Eschelmuller et al., 2025). As in the passive condition, concurrent agonist–antagonist vibration has been found to have variable effects. Eschelmuller et al. (2025) demonstrated that concurrent vibration produces an effect approximately equal to the sum of their individual effects, with no increase in variable error, congruent with the notion of limb motion being inferred from the combined signalling from all muscles acting on the relevant joint. Bock et al. (2007), on the contrary, found increased errors in angle matching and force production during concurrent vibration, and argued for a general degradation of proprioceptive feedback. However

as they only reported absolute errors, it is not clear to what extent these errors in angle matching and force production arise from decreased mean accuracy in a particular direction, or are due to increases in variable errors, making their results more difficult to place in context. Furthermore for the angle-matching task, it is not specified how, or whether, muscle thixotropic effects were accounted for during the passive movement of the left arm (which the right arm was supposed to match), leaving open the possibility of bias in the perceived position (Proske & Gandevia, 2012).

These behavioural findings align with recordings of type Ia afferent activity in the cat and humans. At rest, type Ia afferents exhibit minimal firing rates, with firing rate increasing proportionally to the velocity of muscle lengthening (Matthews, 1963; Roll & Vedel, 1982; Roll et al., 1989). During muscle shortening in the contracting muscle the firing rate decreases below baseline, potentially going completely silent (Macefield & Knellwolf, 2018; Roll & Vedel, 1982). The already low baseline activity creates a natural limit on both the amount and precision of information that can be conveyed through further rate reduction. Consequently with its wider dynamic range of firing rates, the lengthening muscle has been assumed to function as the primary source of joint position/velocity information during active movement, explaining the previously observed limited effects from vibration of the agonist muscle (Capaday & Cooke, 1981).

Electromyographic (EMG) recordings during active movements have shown changes in muscle activity with onset latencies of 40–60 ms (Capaday & Cooke, 1983; Cody et al., 1990). This suggests that a medium-short feedback loop is involved in controlling the relevant muscles in response to Ia afferent feedback; definitely longer than the simplest spinal reflexes, yet well below the time required for conscious and deliberate control. However no similar latency analysis has been conducted to determine the time course of actual movement speed correction.

It should be emphasised that there is, in practice, significant functional overlap between the type Ia and type II afferents, such that they both encode position- and velocity-dependent signals to some extent, but with different primary functions. Their encoding of movement further depends on the activity of the parent muscle and any external load against which it is working; for example acting against gravity may phase shift muscle afferents towards signalling acceleration and velocity, respectively (Banks et al., 2021; Dimitriou & Edin, 2008; Macefield & Knellwolf, 2018). More recent work has indicated the possibility of muscle spindles functioning as peripheral processing units during active movements (Dimitriou, 2022), and that their signalling may be more closely correlated with muscle force, and its time derivatives (Blum et al., 2017, 2020). Together such findings may indicate a split in function between the

passively lengthening antagonist muscle spindle afferents, whose activity appears to signal muscle length and stretch velocity and/or force (and its time-derivatives), while the afferents found in the actively contracting agonist muscle may be involved in processing that depends on the descending drive which activates the intrafusal fibres, facilitating motor control.

In addition to vibration-based approaches visual feedback has a long history of being used to induce perceptual offsets, such as seen in the classical rubber hand illusion (Botvinick & Cohen, 1998), and for redirected reaching tasks (Körding & Wolpert, 2004). It is generally seen that when vision and proprioception are incongruent, vision becomes the primary influence on the perceived position and movement of the hand (Fournerett & Jeannerod, 1997). Virtual reality (VR) presented via a tracked head-mounted stereoscopic display is a clean solution to experimentally manipulate the visual feedback of research participants. This method allows for visual feedback of the hand to be displayed at the tracked position of the hand, while keeping the normal proprioceptive-to-visual mapping, including depth cues (Slater et al., 2008).

In summary, previous research has established that type Ia afferents encode muscle stretch velocity, whereas type II afferent signalling is more directly associated with static muscle length, although their function overlaps to some extent; both position and velocity-related signals can be extracted from both afferent types. The behavioural responses to muscle vibration demonstrate that both perceived limb velocity and position are strongly affected; whereas biased type Ia afferent activity is thought to cause the bulk of the perceptual effects, both type II and Golgi tendon afferents may also contribute to some extent. As such, it remains unknown to what extent proprioceptive inference is cross-modal, such that velocity signals contribute to the perceived position during movement through an integration-like function, thereby providing a signal regarding change in position. This kind of cross-modal sensory inference is consistent with predictive frameworks of sensorimotor control, such as Active Inference and Optimal Feedback Control. Although these frameworks differ in their formulation, both assume that position and velocity signals derive from a shared underlying state and must remain congruent, for example, through smoothness priors placed on higher-order time derivatives (Friston et al., 2010). For example if the arm is perceived to be stationary (zero velocity), then its position should not be changing; conversely, a change in position implies a corresponding velocity. This internal consistency is a basic requirement of any dynamic state estimator and is reflected in these frameworks (Friston et al., 2010; Todorov & Jordan, 2002; Wolpert et al., 1995). Despite these theoretical predictions the extent to which

proprioceptive signals are integrated in such a cross-modal fashion remains empirically unexplored.

In this paper we investigate how sensed limb position and limb velocity are integrated into a cohesive perception of limb state and movement. Our study is structured around three interrelated research questions:

(1) How are changes in perceived and actual movement speed related?
(2) How does muscle vibration affect endpoint errors, movement speed and movement time?
(3) What is the latency of movement correction in response to muscle vibration?

Addressing these questions will enhance our understanding of how positional and velocity signals are integrated for limb state estimation during active movements. To explore these questions we designed three VR-based tasks, which are used in conjunction with muscle vibration.

## Methods

### Ethical approval

The experiment was conducted at the Department of Psychology, University of Copenhagen, and was approved by the Department of Psychology Ethics Committee (no.: IP-EC-030723-01), in accordance with the Declaration of Helsinki (except for pre-registration in a database). Seventeen healthy participants (3 men, 14 women, all right-handed) were included, with a mean age of 23.4 years (range 20–31 years). All participants provided informed written consent before participating in the experiment and were compensated monetarily for their time at a rate of 160 DKK per hour.

The inclusion criteria stated that participants should be healthy, in the age range of 18–50 and be right-handed.

### General experimental procedures

All data were collected in a single session lasting approximately 2 h, sharing the common set-up seen in Fig. 1. The results from the three tasks that will be presented here were collected as part of a larger dataset (Mortensen & Christensen, 2025a), including a total of seven tasks, designed to investigate various aspects of sensory inference and motor control (see the Data Availability Statement section).

Participants were seated and had their right arm placed on a rotating armrest attached to a table, and were outfitted with a VR headset and vibration motors (all described in detail in the sections below). With the arm resting on the rotating armrest and holding the attached right VR controller, participants could freely move their arm in the extension and flexion directions with minimal friction. Trials in all three tasks were initiated by moving the controller to an indicated start position visible in VR and physically close to the chest, with the right elbow flexed. Here participants would press a thumb button on the VR controller to indicate their readiness to start, which would change the colour of a target presented in VR (semi-transparent sphere with a radius of 2.5 cm) from red to yellow. After a short delay (500–1000 ms), the target would turn green, which served as the 'go' signal. Up to this time, a virtual representation of the rotating armrest was visible in the VR environment and was mapped to the actual armrest location. Before each of the three tasks, 10 practice trials were performed.

The target was always presented along the perimeter of a circle, with the centre at the axis of rotation of the rotating armrest, and the radius was adjusted to match the individualised position of the controller on the armrest, according to the individual forearm length. This ensured that the target position could always be reached by the controller by rotating the armrest.

**Task 1.** During Task 1, participants were presented with a target per trial, positioned along a 40°–80° arc in the extension direction from their starting position. The virtual arm remained visible throughout the movement, but its movement speed was pseudo-randomised to one of five conditions:

- 12% slower, 6% slower, True speed, 6% faster or 12% faster

The position of the virtual arm would match that of the real arm at the start of each trial, with a cumulative spatial discrepancy developing over the course of the movement for each of the four manipulated conditions. The virtual arm was hidden during the return movement between each trial. Participants simply needed to move the virtual arm past the target location to end each trial, and were not instructed to move at any particular speed. They were informed that the virtual arm was manipulated to always be slightly slower or faster than the actual speed of their arm (they were not informed of the 'True speed' control condition), and they were instructed to judge whether the movement speed of the virtual arm was 'Faster' or 'Slower' than their actual arm after each trial. The above set-up was combined with the following four vibration conditions, which started upon arm movement and lasted until the target was reached:

- Agonist vibration, Antagonist vibration, Double vibration or No vibration

Each visual speed gain and vibration condition was repeated 18 times for a total of 360 trials per participant, split across two runs to allow for a break in between, in

a pseudo-randomised order, with a uniform distribution of target distances (40°–80°). The goal of this task was to investigate changes in both actual and perceived movement speed in response to muscle vibration. To the extent that muscle vibration affects the perceived movement speed this should be reflected in changes in the distribution of reports of the virtual arm's relative speed.

**Task 2.** During Task 2, targets were presented 40°–80° from the starting position. Participants were instructed to point to the location of the target by moving their arm and aligning their thumb to it before pressing the thumb button on the VR controller. They were not instructed to move at any particular speed and were allowed to make corrections until the thumb button was pressed,

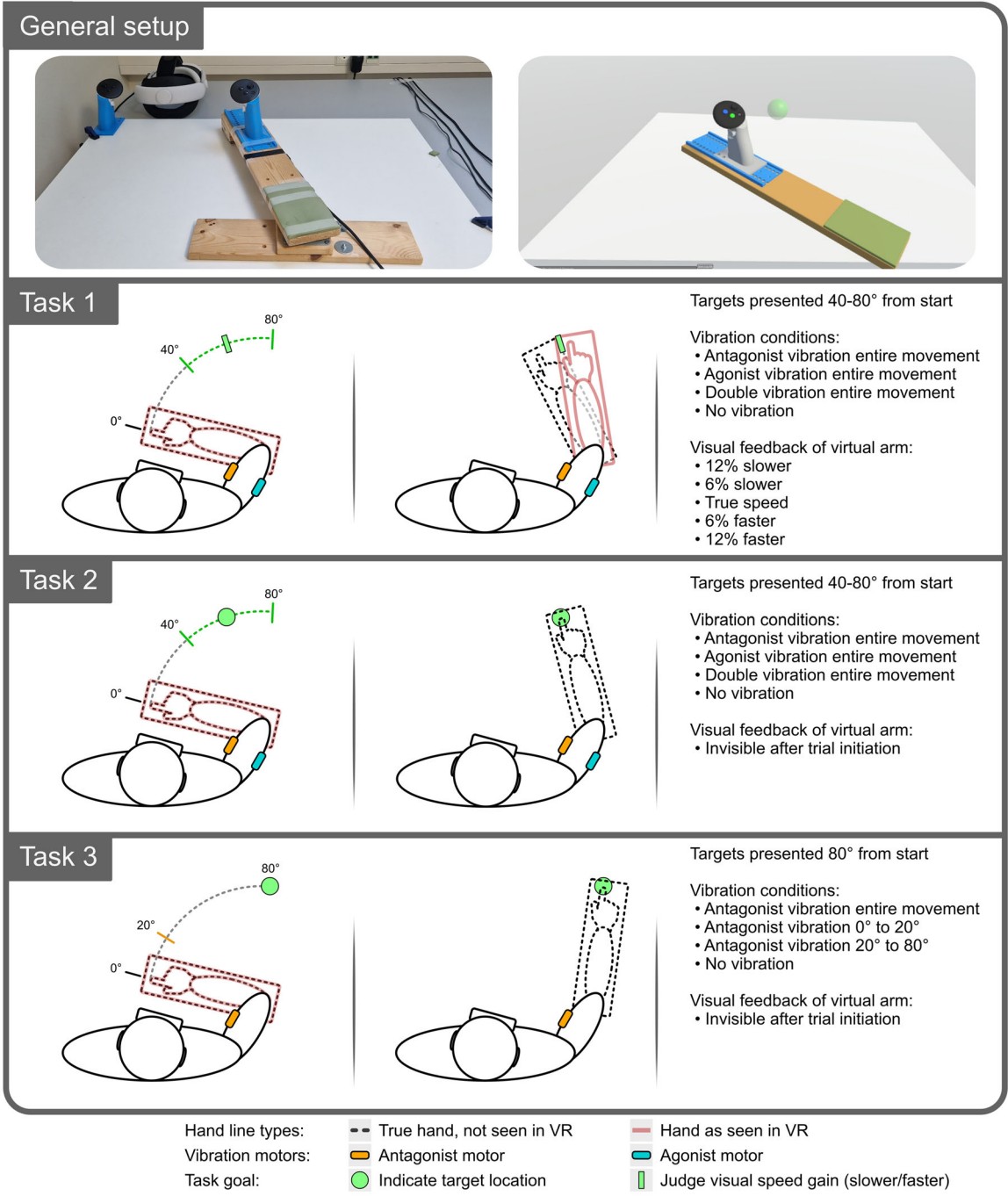

**Figure 1. Experiment overview**
The top panel shows the physical set-up on the left and the virtual reality (VR) representation that subjects see on the right, here including an example target (green sphere). The remaining three panels summarise the experimental set-up.

which ended the trial. They were provided with no feedback regarding their performance. Upon the 'go' signal (target turning green), the virtual arm became invisible, leaving only the tabletop and the target visible in the VR environment. Upon movement onset, one of four vibration conditions was started and lasted until the target location was indicated:

- Agonist vibration, Antagonist vibration, Double vibration or No vibration

The order of vibration conditions was pseudo-randomised, ensuring 24 repetitions of each condition for a total of 96 trials, with a uniform spread of distances to the target. This task was repeated in the flexion direction, starting with the right arm at close to full extension and moving towards the chest for 96 more trials. Our goal with this task was to investigate the effects of muscle vibration on movement speed, time and endpoint errors.

**Task 3.** During Task 3, participants were again instructed to indicate the target position as precisely as possible by aligning their thumb to the perceived position of the target, which was always presented 80° from the start location. Participants were not instructed to move at any particular speed. At the start of the trial the virtual arm would again turn invisible, as in Task 2. To investigate online changes in movement speed in response to turning the antagonist vibration motor on or off during movement, four conditions were used. These are in two sets of intervention-control pairs:

- Vibration 0°–20° compared to Vibration 0°–80°
- Vibration 20°–80° compared to No vibration

We chose to target vibration start/stop at 20° into the movement, to trigger the vibration motor to turn on or switch off early enough that any effects on movement velocity would have time to unfold well before the participant ended the trial.

Forty-eight trials of each condition were completed in a pseudo-randomised order, totalling 192 trials per participant.

### VR set-up

The experiment was programmed in the Unity engine (v21.3.24f1, Unity Technologies, San Francisco, CA, USA), utilising the Unity Experiment Framework (Brookes et al., 2020) to control the trial and block structure of the experiment. We used a Meta Quest 3 VR headset (Meta Platforms Technologies Ireland Limited, Dublin, Ireland) connected to the experiment PC via USB Quest Link, allowing the Unity executable to run on the experiment PC and be displayed in the Quest headset.

The headset was set up to run at a refresh rate of 90 Hz. The left controller was mounted at the corner of the tabletop, allowing the table to be tracked and displayed in VR, providing participants with a stable visual reference of the working environment.

### Rotating armrest

An armrest allowing rotation around a vertical axis was constructed to support the participant's arm during the experiment. The armrest was positioned so that it had an axis of rotation under the participant's right elbow joint, with the individualised height adjusted to just below the shoulder, thereby minimising movements of other joints. A swivel ball bearing was used for the rotation axis. This, along with a smooth tabletop combined with felt sliders on the underside of the armrest end, allowed for very low-friction movement. The right-hand VR controller was mounted on the armrest, with its exact positioning individualised based on forearm length. The controller provided the necessary button inputs for the experiment and real-time tracking of the rotating armrest. See Fig. 1 for visualisation.

The controller mounted on the rotating armrest was tracked via the native inside-out tracking of the Meta Quest 3 headset, with the position of the controller reported at 90 Hz. These tracking data were used when visual feedback of the controller location was used in the experiment (Task 1), as well as for our data analysis. As we are not aware of any specific investigations of the tracking performance and characteristics of the Meta Quest 3 controllers, we evaluated them as follows.

To track the changes in the elbow joint angle, the axis of rotation is required, but it is not directly tracked. We instead estimated it by sampling multiple points along the circle's periphery. During each trial the centre of rotation was re-estimated from 30 equally distributed tracked points along the circle periphery from the previous trial, captured as the participant rotated the armrest from the starting position to the target position. This point was re-estimated in each trial to guard against any potential tracking drift, or if the table itself was nudged slightly during a task, moving the actual axis of rotation. This method of estimating the axis of rotation provided a fairly stable estimate of the axis in the tracked space, with a trial-to-trial SD of 1.2, 0.63 and 0.85 mm in the x (lateral), y (vertical) and z (sagittal) directions, respectively. Some of the stability comes from averaging the circle centre calculated from across 30 points along the perimeter. We further checked the mean within-participant trial-to-trial SD of the estimated distance from this circle centre to the controller, with the tracked controller location taken from a single sample in each trial, to give some measure of reliability when not averaging across tracking points;

as the controller was hard-mounted to the armrest this distance is constant for a given participant. This provided a mean within-participant trial-to-trial SD of 0.29 mm distance. The group mean distance was estimated at 36.1 cm (range 31.6–42.8 cm), corresponding to the elbow to mid-hand distance. Although this is not a measure of the true tracking error (i.e. there could be consistent biases in specific directions) it nevertheless highlights that the tracking solution is quite repeatable and stable in performance.

## Vibration motors

Two small vibration motors (Vp216 VIBRO transducer, Acouve Laboratory Inc., Tokyo, Japan) were mounted on the right arm, with an inelastic micropore tape. This is a coin-shaped voice-coil type vibration motor, with a sprung oscillating mass, driven by an amplifier, similar to loudspeakers. They have a weight of 49 g, a diameter of 43 mm and a height of 15 mm. When mounted flat on the arm the vibration direction is in the normal direction from the skin surface, providing a pushing force directly downwards during vibration. The vibration amplitude was not measured during the experiment and must be expected to continually change somewhat during movement due to changes in mounting pressure, as well as from participant to participant; in positions where the tape holds the motor more firmly against the skin the effectively vibrated mass increases, leading to decreased vibration amplitude of the motor itself. This is a distinct disadvantage compared to larger externally hard-mounted vibration motors; instead they allow for greater and more comfortable movement.

The vibration motors were controlled by an 80 Hz sine wave audio signal produced by the experiment PC's sound card, with the signal amplified by a stereo PAM8610 amplifier (Diodes Incorporated, Plano, TX, USA) using a 12 V 1.5 A power supply. The delay from triggering vibration in the experiment software to actual vibration start was determined (118.8 ms (SD = 4.6)) using an Arduino Uno R3 (Arduino S.r.l., Monza, Italy) analogue-to-digital listener, and was accounted for in the timing-sensitive analysis of Task 3.

One vibration motor was mounted over the distal triceps tendon, just proximal to the elbow, whereas the other was mounted centrally on the belly of the biceps muscle. Although vibration experiments most often target the muscle tendons rather than the muscle belly directly, previous research has indicated no difference in the required vibration amplitude to elicit an illusory movement between the distal biceps tendon and the biceps muscle belly (Ferrari et al., 2019).

The mounting of both motors was chosen to minimise pressure changes during elbow flexion/extension; with the motors mounted to the arm via tape, the muscles are able to slide under them, substantially altering the circumference of the arm at the mounting point, if the muscle is sometimes under the motor, and sometimes not.

We found the best compromise here was to mount the biceps motor centrally on the muscle belly, such that the biceps muscle would never shorten enough to no longer be under the motor. The triceps motor was instead mounted over the distal tendon, such that the triceps would never lengthen enough to slide under the motor. We were unable to find a pair of mounting spots where both motors would consistently target either the muscle belly or tendon throughout the whole flexion-extension range. Pilot testing indicated that the chosen mounting spots nevertheless produced the expected proprioceptive illusions in both flexion and extension movements.

## Data processing and statistical modelling

Data analysis was conducted in R (v4.3.3, (R Core Team, 2024)) with data wrangling and plotting completed using the tidyverse package (v2.0.0, (Wickham et al., 2019)). Model fitting was performed using the brms package (v2.20.4, (Bürkner, 2017)), an R interface for the probabilistic programming language Stan (v2.32.6, (Stan Development Team, 2024)). In addition the PyMC library (v5.10.2, (Abril-Pla et al., 2023)), in Python (v3.7.11, (Python Software Foundation, 2023)) was also used. All data analysis code is available in the code repository (Mortensen & Christensen, 2025b); see the Data Availability Statement section.

We chose to use Bayesian regression modelling to analyse our data, as we believe it provides a more straightforward interpretation of parameter estimates, compared to analogous frequentist models. Our models were specified so as to provide directly interpretable parameter estimates. In addition, the Bayesian framework naturally supports partial pooling, which appropriately accounts for the nested structure of trials within participants; individuals must be expected to show somewhat variable responses to, for example, muscle vibration. We present the regression models here in a minimal form, focusing on readability and omitting the specification of the nested structure. For all models we included individual-level intercepts and slopes for all the same group-level parameters that are presented here. Full model specifications and priors are available in the code repository.

All models used weakly informative priors with effect parameters centred at zero, providing mild regularisation while allowing the data to dominate the posterior estimates; that is, *a priori* the most probable effect (e.g. of muscle vibration) is assumed to be zero, to ensure that any estimated effect is driven by the data, rather than the specified prior. Inference was

performed using the No-U-Turn Sampler, a variant of Hamiltonian Monte Carlo. All models were run with four independent chains of 4000 iterations each, including 1000 warm-up iterations, yielding 12,000 post-warm-up samples. Convergence diagnostics indicated no divergent transitions, all $\hat{R}$ values were below 1.005 and visual inspection of caterpillar plots indicated good convergence and mixing.

Posterior distributions for relevant parameters are summarised using the maximum *a posteriori* (MAP) estimate and the associated 95% credible intervals (CI). An effect was interpreted as supported when the 95% CI did not include zero. All fitted traces, including complete sets of posterior distributions, are made available in the code repository (see the Data Availability Statement section).

All position and velocity data are presented as angular values of the elbow joint.

**Task 1.** In Task 1, we investigated the effect of muscle vibration on the perceived and actual movement speed. Both models for these two analyses were implemented in brms as hierarchical (multilevel) models with the random effects defined at the participant level.

Our goal with the first model (eqn (1)) was to investigate whether the vibration motors influence perceived movement speed; we modelled the distribution of ratings of whether the virtual arm was rated to be 'Faster' or 'Slower' than the participant's true arm. Specifically, our hypothesis here was that muscle vibration may change the perception of the movement speed of the executed elbow extension movement; for example, antagonist vibration was expected to increase the perceived velocity of biceps muscle stretch. We probe this perception of movement speed by having the participant report the relative movement speed of the virtual arm. If the antagonist muscle vibration increases the proprioceptively signalled movement speed, we should expect to see an increase in the probability of rating the virtual arm as slower. By including the visual speed gain applied to the virtual arm as a predictor in the model, we can confirm whether the participants answer as expected; when the virtual arm is actually moving faster, we should expect a higher proportion of 'Faster' ratings reflecting this. This also establishes a scale based on known movement speed changes, which the parameter estimates of the muscle vibration predictors can be judged against.

$$\text{rating}_i \sim \text{Bernoulli}\,(\Phi\,(z_i))$$

$$z_i = \beta_0 + \beta_1 \text{vg}_i + \beta_2 \text{agovib}_i + \beta_3 \text{antavib}_i \quad (1)$$

We modelled this as a Bernoulli model with a probit link, with the visual speed gain (vg), agonist vibration (agovib) and antagonist vibration (antavib) as predictors.

This model assumes that the answer distribution for each condition can be approximated by a latent, z-normalised Gaussian distribution ($z_i^*$ below), with a static threshold at zero defining whether 'Faster' or 'Slower' is selected:

$$z_i^* = \beta_0 + \beta_1 \text{vg}_i + \beta_2 \text{agovib}_i + \beta_3 \text{antavib}_i + \varepsilon_i$$

$$\varepsilon_i \sim \mathcal{N}\,(0, 1)$$

$$\text{rating}_i = \begin{cases} \text{'Faster'}, & \text{if } z_i^* > 0 \\ \text{'Slower'}, & \text{if } z_i^* \leq 0 \end{cases}$$

This latent distribution can be thought to represent the result of an internal comparison between the proprioceptively perceived speed of the actual arm and the visually observed speed of the virtual arm. On any particular trial, both the proprioceptively perceived speed and visually observed speed may be either over- or under-estimated due to sensory noise, but across trials we should expect a subject with no bias from either proprioception or vision to report the virtual arm as 'Faster' on approximately 50% of trials, when no vibration or visual speed gain is applied; this corresponds to an expected estimate of the intercept ($\beta_0$) of around 0 ($z_i^*$ would then be equally distributed around the threshold at 0). The parameter estimate of, for example, $\beta_3$ then describes how mu of the latent Gaussian is shifted relative to $\beta_0$ when antagonist vibration is applied. Note that visual speed gain is included as a linear predictor on the latent scale, such that a +12% visual speed gain is expected to have twice the effect of a +6% gain.

This type of model is closely related to classical Signal Detection Theory models (Stanislaw, 1999), with the conceptual difference that the signal here (relative speed of the virtual arm) may be either higher or lower than the reference value of 0.

We additionally modelled any changes in trial-wise mean movement speed (mms, in °/s) in eqn (2). For each participant mms was centred relative to the mean in the no-vibration/normal-visual-feedback condition, because we are primarily interested in changes in speed due to muscle vibration rather than baseline differences between participants. Our goal with this model was specifically to investigate whether muscle vibration causes any changes in actual movement, which complements our above model, which examined the effects on the perceived movement speed.

Target distance (td) is included here as a predictor because it strongly predicts variability in movement speed; including td thus reduces unexplained variance and thereby improves the precision of estimates of the vibration and visual-gain effects.

$$\text{mms}_i \sim \mathcal{N}\,(z_i, \sigma)$$

$$z_i = \beta_0 + \beta_1 \text{vg}_i + \beta_2 \text{agovib}_i + \beta_3 \text{antavib}_i$$

$$+ \beta_4 \text{td}_i \quad (2)$$

A total of 360 trials were completed per participant, except for one, for whom the last 194 trials could not be completed due to technical issues. This left a total of 5926 included trials for both models.

**Task 2.** In eqn (3) we specified the model to investigate how endpoint errors, mean movement speed and movement time are jointly affected by muscle vibration. These three outcomes describe different aspects of the performed trial, but are also clearly interrelated; if, for example, it is observed that antagonist muscle vibration leads to undershooting of the target, we should likewise expect to see this reflected in a concurrent reduction in movement speed, a decrease in movement time or some combination of the two. By including all three outcomes in a single model we can investigate which aspects of the movement are affected by the experimental conditions and how the residuals correlate after accounting for these effects.

As such we modelled endpoint errors (epe, in °), mean movement speed (mms, in °/s) and movement time (mt, in s) during Task 2 as outcomes in a hierarchical multivariate model. This was implemented in brms. Before model fitting, the data were centred by the mean in the no-vibration condition within-participant, for each outcome. This enabled us to estimate how each vibration condition affects the three outcome variables, relative to the case when no vibration was present; we were here interested in how, for example, endpoint error accuracy changes in response to muscle vibration, while accounting for any individual baseline differences. Apart from the vibration conditions (agovib and antavib) we also included the distance to the target (td), to account for baseline differences in endpoint errors, movement speed and movement time that depend on the distance to the target. In addition the movement direction (md) and its interactions with muscle vibration and target distance variables were included to differentiate between effects observed for flexion and extension direction movements. This also accounts for the fact that the biceps functioned as the antagonist during extension but as the agonist during flexion, and vice versa for the triceps.

An interaction term between the two muscle vibrations was also included here, specifically to investigate the extent to which the combination of both vibration motors has different effects than the sum of the effects of each type of muscle vibration alone.

$$
y_i = \begin{bmatrix} \text{epe}_i \\ \text{mms}_i \\ \text{mt}_i \end{bmatrix} \sim \mathcal{N}(\mathbf{z}_i, \Sigma), \quad z_i = \begin{bmatrix} z_i^{(\text{epe})} \\ z_i^{(\text{mms})} \\ z_i^{(\text{mt})} \end{bmatrix}
$$

$$
z_i^{(k)} = \beta_0^{(k)} + \beta_1^{(k)} (\text{agovib}_i \times \text{antavib}_i \times \text{md}_i)
$$
$$
+ \beta_2^{(k)} (\text{td}_i \times \text{md}_i)
$$

$$
\sigma_i^{(k)} = f(\text{agovib}_i \times \text{antavib}_i \times \text{md}_i)
$$
$$
\text{for } k \in \{\text{epe, mms, mt}\} \tag{3}
$$

Here $y_i$ contains three observations from each trial, which we model jointly as a multivariate Gaussian. This allowed us to capture shared variability across the three response measures. Interaction terms are expanded in the standard way, such that all main effects and two- and three-way interactions among agovib, antavib, md and td were included as predictors ($a \times b \times c = a + b + c + ab + ac + bc + abc$). Each outcome's residual SD, ($\sigma_i^{(k)}$), was modelled as a function of the vibration and movement direction factors, allowing residual variability to differ across experimental conditions. This allowed us to investigate whether either vibration motor alone or both together had any effect on the spread of variable errors, as would be expected if there is some general degradation of proprioceptive feedback rather than just the induction of a bias in a particular direction. The residual covariance $\Sigma$ is unrestricted, allowing us to estimate correlations between outcomes after accounting for the effects of the modelled predictors.

A total of 192 trials (96 per direction) were included per participant, for a total of 3264.

**Integrative analysis of Tasks 1 and 2.** A further analysis of Tasks 1 and 2 was completed, using only the baseline control trials from each task, that is, a subset of the data also used for the analyses described in the above two sections. Specifically trials with no visual speed gain and no vibration were included from Task 1, whereas trials with no vibration were included from Task 2.

The first model here shares the same Bernoulli (probit-link) structure as the first model described for Task 1; however here we examine whether mean movement speed (mms) predicts the tendency to overestimate or underestimate perceived movement speed. To account for between-participant differences in both the mean and variability of movement speed, mms was z-normalised within-participant before model fitting (denoted as $\text{mms}_{z\text{-norm}}$). Our goal with this model was to investigate whether within-participant variation in movement speed is predictive of the ratings of the relative movement speed of the virtual arm.

$$
\text{rating}_i \sim \text{Bernoulli}(\Phi(z_i))
$$
$$
z_i = \beta_0 + \beta_1 \text{mms}_{z\text{-norm},i} + \beta_2 \text{td}_i \tag{4}
$$

A total of 18 trials were included per participant for a total of 296 trials (only eight trials were included for one participant, as mentioned in the Task 1 section).

The second model (eqn (5)) is set up to investigate if and to what extent movement speed is predictive of endpoint errors. Together eqns (4) and (5) were set up

to complement each other, such that we may investigate both how perceived and actual movement speed covary, as well as how movement speed and endpoint errors covary; by testing these in only the baseline trials with no induced perturbations we can infer to what extent the effects observed in the main analyses of Tasks 1 and 2 represent interrelationships between perceived and actual movement speed and perceived position that are fundamental to sensory inference, or whether they are multiple independent effects, all provoked by muscle vibration.

Both the endpoint errors and mean movement speed are likewise z-normalised ($epe_{z\text{-norm}}$ and $mms_{z\text{-norm}}$) to account for between-subject differences as above.

$$epe_{z\text{-norm},i} \sim \mathcal{N}(z_i, \sigma)$$

$$z_i = \beta_0 + \beta_1 \left(mms_{z\text{-norm},i} \times md_i\right)$$
$$+ \beta_2 \left(td_i \times md_i\right) \tag{5}$$

Target distance (td) is included in both models to capture any effect of varied target distances on the respective outcomes. Movement direction (md) and its interactions are included in eqn (5), similarly as described for eqn (3).

A total of 48 trials per participant (24 per direction) were included, for a total of 816 trials.

**Task 3.** Our goal with this task was to investigate the latency of any change in movement speed from when the antagonist vibration motor was turned on or switched off. To investigate this we implemented a changepoint model (e.g. see [Carlin et al., 1992]) in PyMC. To run the model effectively, some preprocessing of the tracking data was required. First, dmi the time was adjusted to be set to 0 at the time of starting/stopping the vibration motor, which was triggered at 20° into the movement. Data from 100 ms before to 120 ms after vibration start/stop were included in the model fit. The raw tracking data were position-based, and to estimate angular movement speed, we used the first derivative of a cubic smoothing spline applied to the angular position data, using R's smooth.spline() function with the smoothing parameter (spar) set to 0.4. The movement speed was extracted at 10 ms intervals, which was then used in the changepoint model as follows.

The mean movement speed at each time point $t_i$ was calculated across trials within each control condition ('No vibration' and 'Antagonist vibration 0°–80°'), within-participant. These mean values were then subtracted from the trial-level movement speed data, at each time point $t_i$, in the corresponding intervention condition ('Vibration 20°–80°' and 'Antagonist vibration 0°–20°'). After this step the data represent the timeline of delta movement speed in each intervention condition trial, relative to the mean movement speed at each time

point in the corresponding control condition for that participant.

The mean of this delta movement speed should now be close to zero until vibration start/stop begins to affect the movement speed in the intervention condition, at which point the delta movement speed should then begin to diverge away. The changepoint model is set up to estimate this time point. To see a visualisation of this, along with the fitting process, see Fig. A2 in the appendix.

For simplicity, only the model likelihood, along with the priors of the changepoint, is shown here in eqn (6). $dms_i$ is the delta movement speed at time $t_i$. This is used to sample the most likely time of change in movement speed (group mean changepoint denoted $\bar{\tau}$, participant-to-participant SD denoted $\tau_\sigma$), given the model formulation below. To allow for sampling of a continuous changepoint we use the value of a sigmoid function ($S_i$ at $t_i$) to blend smoothly between the pre- and post-changepoint linear functions. A uniform prior for the group mean changepoint was used, between 0 ms (vibration start/stop) and +120 ms. The same model, including identical priors, was run twice, once each for the two sets of intervention-control pairs of conditions. The complete model definition with all priors is provided in the appendix (Eq. A1), where it is also described in further detail.

$$dms_i \sim \mathcal{N}\left(\mu_i, \sigma_j\right)$$

$$\mu_i = \mu_{\text{pre}\_\tau,i} \cdot S_i + \mu_{\text{post}\_\tau,i} \cdot (1 - S_i)$$

$$S_i = \frac{1}{1 + e^{c_i}}$$

$$c_i = \frac{t_i - \tau_j}{0.001}$$

$$\mu_{\text{pre}\_\tau,i} = \alpha_{\text{pre}\_\tau,j} + \beta_{\text{pre}\_\tau,j} \cdot t_i$$

$$\mu_{\text{post}\_\tau,i} = \alpha_{\text{post}\_\tau,j} + \beta_{\text{post}\_\tau,j} \cdot t_i$$

$$\tau_j = \bar{\tau} + \tau_\sigma \cdot \varepsilon_{\tau,j}$$

$$\varepsilon_{\tau,j} \sim \mathcal{N}(0, 1)$$

$$\bar{\tau} \sim \mathcal{U}(0, 0.12)$$

$$\tau_\sigma \sim \mathcal{HN}(0, 0.025)$$

$$\text{for } j \in \{1, \dots, n_{\text{id}}\} \tag{6}$$

A total of 48 trials of each of the four conditions per participant (816 per condition) were completed during data collection. However a small number of trials were excluded due to transient tracking issues. A total of 16 trials were discarded across the four conditions. Data from −100 ms before to 120 ms after vibration start/stop were included, at 10 ms intervals for 23 observations per trial.

To test whether the above-mentioned spline method was a reasonable approach for estimating movement speed

for this use case, a simple finite-difference approach to estimating angular movement speed was also tested with the same changepoint model. Here, the movement speed was estimated as follows, before being linearly interpolated to align samples temporally:

$$ms_i = \frac{angle_{i+1} - angle_{i-1}}{t_{i+1} - t_{i-1}}$$

Both methods produced group mean latency estimates within 2 ms of each other for both model fits (vibration on and off), whereas the spline method filtered out some of the noise apparent at the trial level in the finite-differences approach. Both sets of estimated movement speed data and model traces are available in the code and data repository mentioned in the Data Availability section.

## Results

### Task 1

In Task 1, we investigated the effects of muscle vibration on both the perceived and actual movement speed.

As shown in Fig. 2*A*, we found a 51.6% (95% CI [42.1, 60.6]) probability of rating the virtual arm as 'Faster', in the control condition with no vibration and no visual speed gain, indicating that the average estimation of relative movement speed is correct. Varying the visual speed gain from −12% to +12%, shows the expected shift in the probability of reporting the virtual arm as 'Faster'; when the visual speed gain is increased, the probability of reporting it as such increases, and vice versa when it is decreased. This effect was estimated with a linear coefficient of 0.41 (95% CI [0.31, 0.52]) for a visual gain of +6% (purple line interval, Fig. 2*B*). This estimate corresponds to the 'slope' of the black line in Fig. 2*A*; this variable is modelled linearly on the probit latent scale, producing the observed S-shaped curve in the visual-speed-gain-to-probability plot. This leads to a group mean probability of correctly rating the virtual arm as faster of 67.2% (95% CI [56.3, 77.2]) for the +6% condition and 80.2% (95% CI [68.8, 89.2]) for the +12% condition, with similar changes in the opposite direction for the negative speed gains.

Having thus validated that the participants can successfully compare their own arm movement speed to that of the virtual arm, we can interpret the effects of applying muscle vibration during this task. Here we found a decrease in the probability of reporting the virtual arm as 'Faster' when antagonist vibration is applied, with a coefficient of −0.29 (95% CI [−0.47, −0.1]), which corresponds to a decrease to 40.3% (95% CI [31.2, 49.6]) probability of reporting the virtual arm as faster when no visual speed gain is applied. This may then be interpreted as antagonist muscle vibration causing participants to

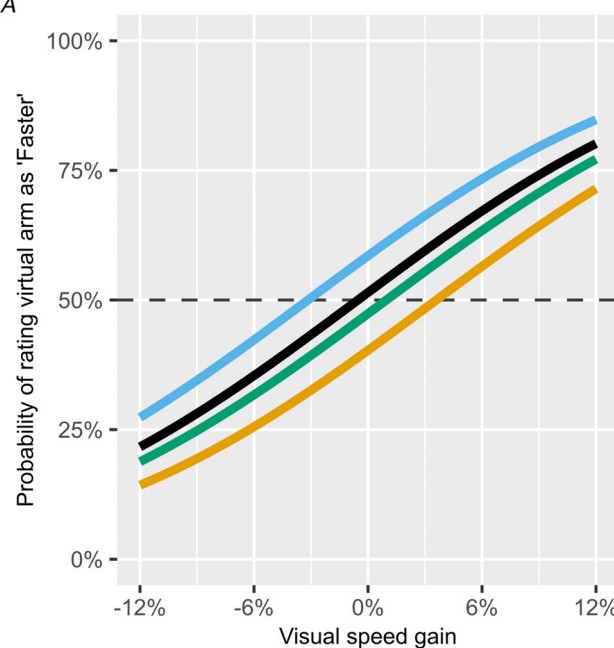

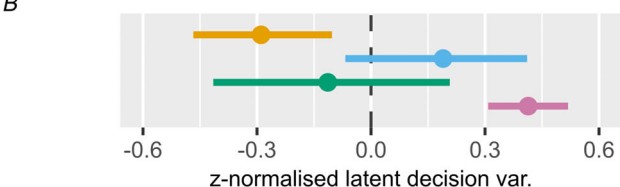

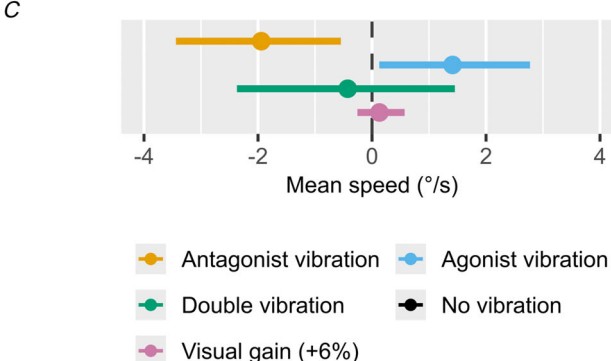

**Figure 2. Task 1 results**
Panel *A* shows the modelled probability of rating visual feedback as appearing 'Faster', for the control 'No vibration' condition (grey), as well as with agonist vibration (teal) and antagonist vibration (orange). Panel *B* shows the modelled z-normalised latent distribution differences between the agonist and antagonist vibration conditions relative to the 'No vibration' condition. As a point of reference, the modelled effect of increased speed of +6% visual speed gain is also shown. Panel *C* shows how the same three conditions affect mean movement speed compared to the 'No vibration, normal visual speed' control condition. Dots and point intervals represent maximum *a posteriori* (MAP) and 95% credible interval (CI). The model results are based on 360 trials per participant (*N* = 17), divided across 18 trials per vibration and visual speed gain conditions. Due to some uncompleted trials for one participant a total of 5926 trials are included.

overestimate their movement speed, as they here become less likely to report the virtual arm as 'Faster'. For further visualisation of the underlying rating data see Fig. A1 in the appendix.

In addition to biasing the perceived movement speed, antagonist vibration also causes adjustments of the actual movement speed; as the participants believe they are moving faster they also decrease their mean movement speed by 1.94°/s (95% CI [0.55, 3.44]).

For agonist vibration the results are mixed. Although the actual movement speed increased by 1.41°/s (95% CI [0.13, 2.88]), the 95% CI describing the change in probability of reporting the perceived movement speed overlapped 0, indicating insufficient evidence to claim such an effect. Concurrent vibration of both motors tended to cancel each other out, such that the perceived relative speed and actual movement speed remained unchanged compared to the no vibration condition.

The effect of visual speed gain on actual movement speed was similarly modelled as a linear predictor, such that the +12% visual speed gain was expected to have twice the effect of +6%, with the negative speed gains producing similar effects in the opposite direction. Here, we specifically found strong support for no practical change in movement speed, with a parameter estimate of 0.17°/s (95% CI [−0.23, 0.6]) for the +6% condition, almost perfectly centred at zero, with a very tight CI. It is worth reiterating here that the participants were explicitly informed that a visual speed gain would be applied to the virtual arm, and that they would be required to rate the relative speed of the virtual arm after each trial. This might have encouraged a strategy where they primarily relied on proprioceptive rather than visual feedback to control the movement.

## Task 2

In Task 2 we investigated how muscle vibration affects movement time, movement speed, and endpoint errors (Fig. 3). Although we focus on the changes in performance caused by muscle vibration in the following section, a summary of the baseline performance in the 'No vibration' trials is provided in Table A1 for reference in the appendix.

We found that antagonist vibration caused a consistent bias towards undershooting the target, both in the flexion (−2.76° (95% CI [−3.8, −1.64])) and extension directions (−1.32° (95% CI [−2.07, −0.48])). Agonist vibration, in contrast, caused participants to overshoot the target in the flexion direction (1.45° (95% CI [0.51, 2.39])), but not in the extension direction. The estimated interaction effect of agonist and antagonist vibration together overlapped zero, indicating that their combined effect is not different from the sum of each effect alone; this produced a resulting undershooting effect of double vibration during flexion,

but not extension movements. Furthermore we found no indication of increased SD of endpoint errors, suggesting that neither agonist, antagonist, nor both of them together have any effects on movement precision (Fig. 3*D*).

Mirroring our observations in Task 1, antagonist vibration caused a decrease in movement speed (flexion direction: −3.13°/s (95% CI [−4.09, −2.05]), extension direction: −1.18°/s (95% CI [−2.02, −0.44])), whereas agonist vibration resulted in an increase in movement speed (flexion direction: 2.71°/s (95% CI [1.84, 3.54]), extension direction: 1.25°/s (95% CI [0.5, 2.05])). Similar to the effects on endpoint errors, the estimated vibration interaction effect indicates that changes in movement speed during double vibration are effectively the sum of changes during agonist and antagonist vibration alone. This resulted in a reduction of movement speed during flexion, but not extension movements, matching the effects observed for changes in endpoint errors.

Additionally, we found small but consistent effects of muscle vibration on movement time, but in the opposite direction compared to what one might expect, considering the described effects on endpoint errors and movement speed. In a simplified manner, a change towards undershooting the target due to antagonist vibration may have been caused by either a decrease in movement time at a similar movement speed or a reduction in movement speed at a similar movement time. In contrast to this, we found that antagonist vibration during flexion movement caused a bias towards increased movement time of 52.48 ms (95% CI [6.34, 99.79]), whereas agonist vibration caused decreases in movement time of −73.9 ms (95% CI [−108.62, −40.57]) for flexion movements and −37.48 ms (95% CI [−65.65, −8.14]) for extension movements. These changes in movement time effectively work in the opposite direction as the changes in movement speed, in terms of achieving a given endpoint error. Here we again did not find any indication of vibration interaction effects.

Finally we found a residual correlation of 0.20 (95% CI [0.17, 0.24], Fig. 3*E*) between movement speed and endpoint errors, indicating that movement speed is predictive of endpoint errors, even after accounting for the modelled effects of the vibration motors on both parameters.

To further highlight how endpoint error and mean movement speed are jointly affected by muscle vibration they are plotted against each other in Fig. 4. In this plot endpoint errors and mean movement speed are both z-normalised by the values in the no vibration condition, within each participant, to account for subject-to-subject variation in both baseline bias and trial-to-trial variation, while highlighting how endpoint errors and mean speed change in response to muscle vibration. Here, it becomes apparent how antagonist vibration pushes the distribution down and to the left, while agonist vibration pushes it up and to the right, with an intermediate effect of double vibration.

## Integrative analysis of Tasks 1 and 2

One way to jointly interpret the above findings from Tasks 1 and 2 is to assume that the CNS estimates limb state in a cross-modal fashion, with position- and velocity-based proprioceptive signals augmenting each other and contributing to a shared state estimate, as in predictive frameworks of sensorimotor control. We further explore this notion by considering the subset of baseline control trials from Tasks 1 and 2 with no vibration and no visual speed gain, as we should expect to see this same effect reflected here, and not only in perturbed trials.

In Task 1 we demonstrated that antagonist vibration causes participants to both overestimate movement speed and slow down their reaching movements, whereas agonist vibration caused a similar effect of increasing movement speed, although we could not show the corresponding perceptual effect here. Although we are

only able to capture the report of the perceived movement speed after each trial is completed with the current experimental design, it seems reasonable to assume that the movement speed is actually changed as a response to the biased feedback that is induced through muscle vibration.

This might indicate that the motor plan that the participants used to complete this task included some intended movement speed profile; as proprioceptive feedback became biased due to muscle vibration shortly after movement onset, the perceived velocity no longer matched the intended and the change in movement speed that we see is a corrective action taken to realign them.

If this is the case, then we should assume that such corrective actions are also taken during baseline trials; due to sensory noise, the perceived velocity will on any given trial be over- or under-estimated to some extent, even if these errors may average out across trials, as seen in the

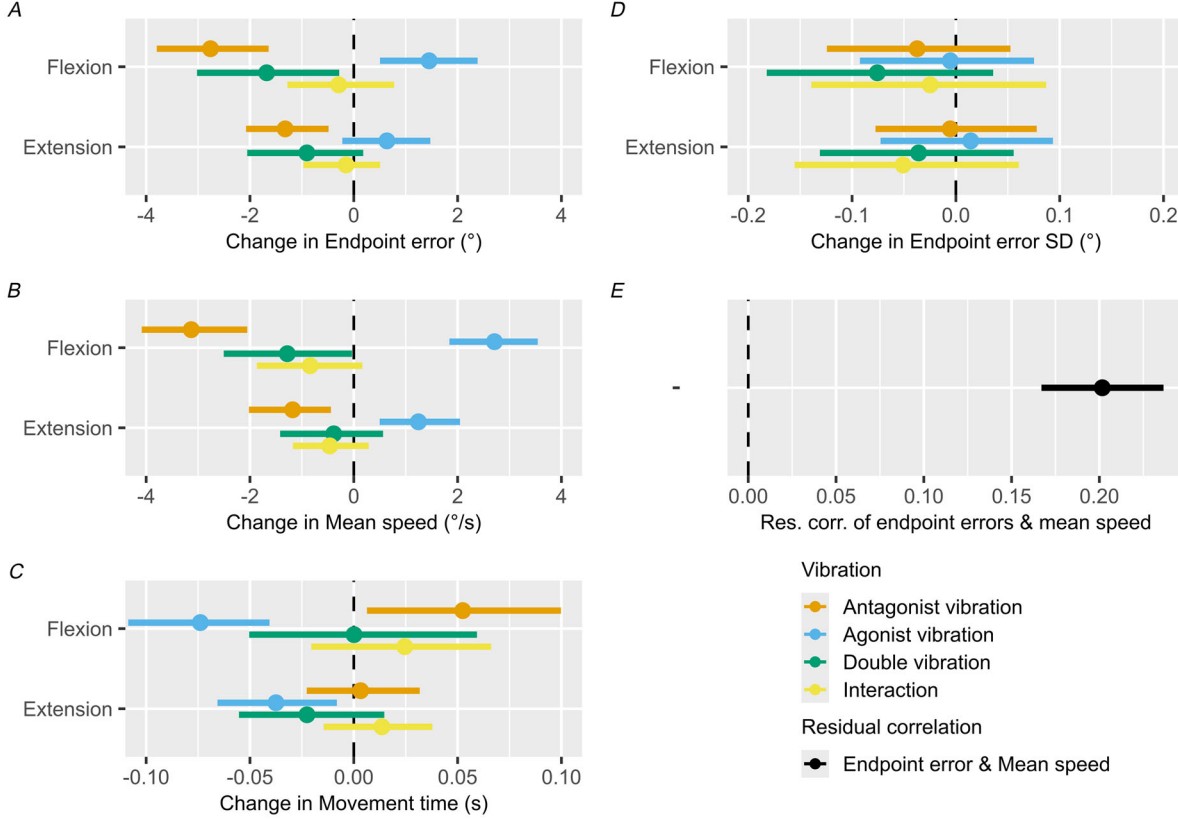

### Figure 3. Task 2 results

Panels *A–C* show the modelled effects of agonist and antagonist vibration, their interaction, and both together. Effects are shown for flexion and extension movements relative to the control 'No vibration' condition, on endpoint errors (*A*), mean movement speed (*B*), and movement time (*C*), respectively. Negative values indicate undershooting the target, decreased mean movement speed, and decreased movement time, relative to the 'No vibration' condition. Effects are shown conditional on the mean target distance of 60° (range 40°–80°). Panel *D* shows the effects on the change in SD of endpoint errors relative to the 'No vibration' condition. Panel *E* shows the modelled residual correlation of endpoint errors and mean movement speed. Dots and point intervals represent maximum *a posteriori* (MAP) and 95% credible interval (CI). The model results are based on 24 trials per vibration condition (no vibration, antagonist vibration, agonist vibration, and double vibration) in both the extension and flexion directions, leading to 192 trials per participant (*N* = 17), and 3264 trials in total.

results of Task 1. As such, while there may be a multitude of reasons for variable movement speed trial-to-trial, one may be that the movement speed was mis-estimated, causing a corrective slowdown or speedup.

This is exactly what we see in Fig. 5*A*, where we demonstrate that z-normalised mean movement speed is predictive of the perceived movement speed, with an estimated coefficient of 0.37 (95% CI [0.14, 0.61]) per SD of z-normalised movement speed; specifically, a trial where a participant moved faster than their own average movement speed is predictive of the participant having underestimated their movement speed, and vice versa.

This thus expands on the results of Task 1, where antagonist vibration concurrently induced both a decrease in actual movement speed and an increase in the perceived movement speed. Although from these results alone it would be difficult to say to what extent these two

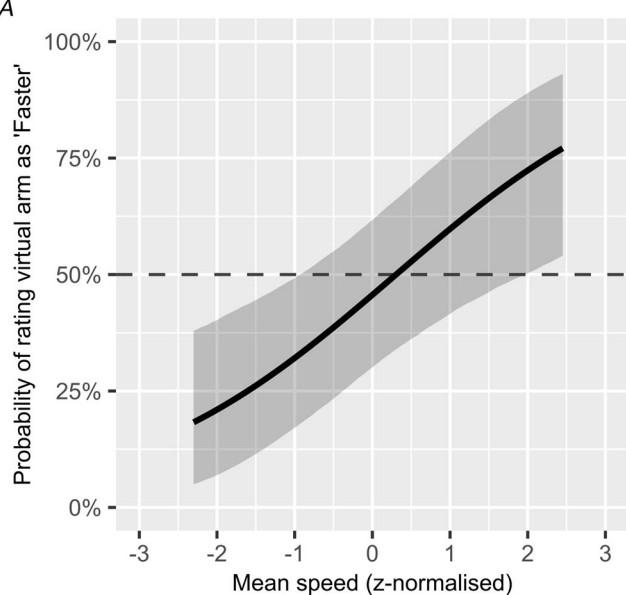

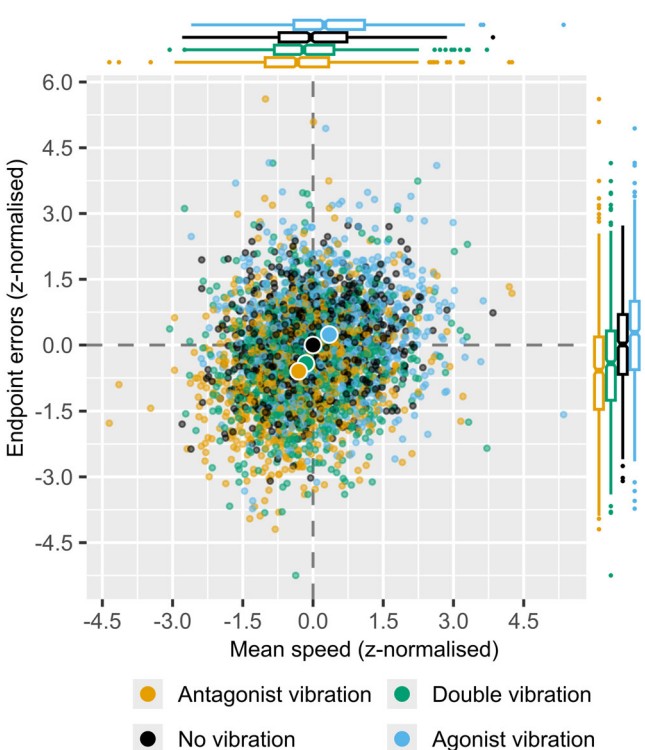

**Figure 4. Z-normalised endpoint errors by mean movement speed**
This plot shows mean movement speed and endpoint errors from Task 2. All values are z-normalised within subject by the values of the control 'No vibration' condition, to account for subject-to-subject variation in both movement speed and endpoint error mean and trial-to-trial variation. Small dots represent individual trials, whereas large dots show the mean of each vibration condition. Boxplots show the distributions of the observed values along each axis. Both flexion and extension direction trials are included. The plot is based on 24 trials per vibration condition (no vibration, antagonist vibration, agonist vibration, and double vibration) in both the extension and flexion directions, leading to 192 trials per participant (*N* = 17), and 3264 trials in total.

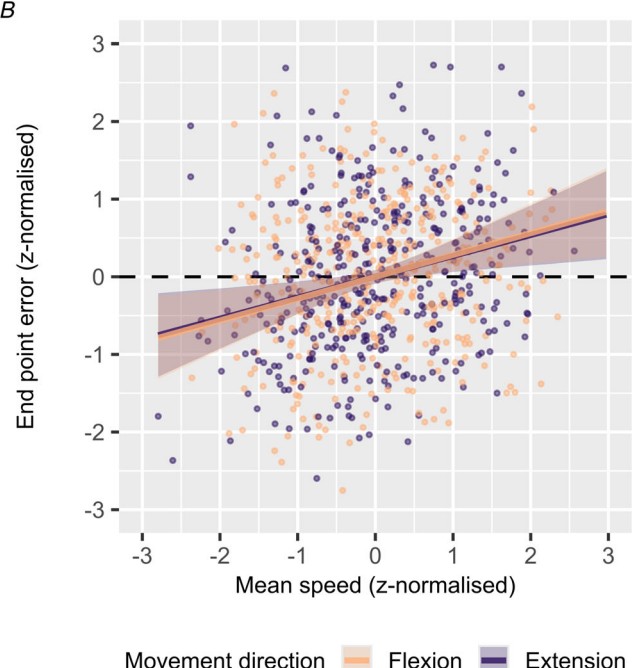

**Figure 5. Task 1 and 2 baseline trials**
Panel *A* shows the maximum *a posteriori* (MAP) and 95% credible interval (CI) predicted probability of rating virtual arm as 'Faster' in the subset of trials from Task 1 with 'No vibration & no visual speed gain', dependent on z-normalised mean movement speed. These model results are based on 18 trials per participant (*N* = 17), for a total 296 trials (10 trials incomplete for one participant). Panel *B* shows z-normalised (within-subject and direction) mean speed and endpoint errors from the subset of trials with 'No vibration' from Task 2, for both flexion and extension direction trials. Dots show individual trials, with overlaid posterior conditional effects linear regression lines showing the MAP and 95% CI. These results are based on 24 trials per movement direction per participant (*N* = 17), for a total of 816 trials.

effects are related, here we clearly see the same negative correlation between perceived and actual movement in non-perturbed trials. This would tend to support the notion that muscle vibration has an effect on the sensed movement speed, which leads to a corrective speedup or slowdown, and further demonstrates that some of the variability in observed movement speed likely arises as a downstream effect of mis-estimated movement speed, whether induced by an external perturbation or not. This further allows us to expand on the interpretation of the results of Task 2, where we observed the same changes in movement speed in response to muscle vibration, along with concurrent changes in endpoint errors.

The changes in endpoint errors in Task 2 demonstrate systematic changes in the perception of travelled arm distance and hand position at the end of each reach; these induced errors in perceived position can now be linked to errors in the perception of movement speed. To further explore this notion we analysed the subset of trials from Task 2 with no vibration, with the expectation that we should see variable endpoint errors and mean movement speed to be correlated. The results of this analysis are shown in Fig. 5*B*, where we indeed found z-normalised movement speed to be predictive of z-normalised end-

point errors, with practically identical estimated slopes of 0.27 (95% CI [0.08, 0.46]) for extension movements and 0.27 (95% CI [0.09, 0.47]) for flexion movements. These results are consistent with the estimated residual correlation between endpoint errors and movement speed shown in Fig. 3*E*.

From Fig. 5*B* it is clear that there is substantial uncorrelated variability in movement speed and endpoint errors; we do not here suggest that all inferred position information originates from velocity feedback sources, but rather that inferred velocity does appear to contribute meaningfully to the perception of position.

## Task 3

In Task 3, we adjusted the experimental set-up to get a more detailed estimate of how the sensorimotor system incorporates information from the Ia afferents in real-time during movement, by turning on or off the antagonist vibration motor 20° into an 80° extension movement.

Figure 6 shows the same two control conditions in both Panels *A* and *B*, with no vibration in the dashed grey line and antagonist vibration over the entire movement in the

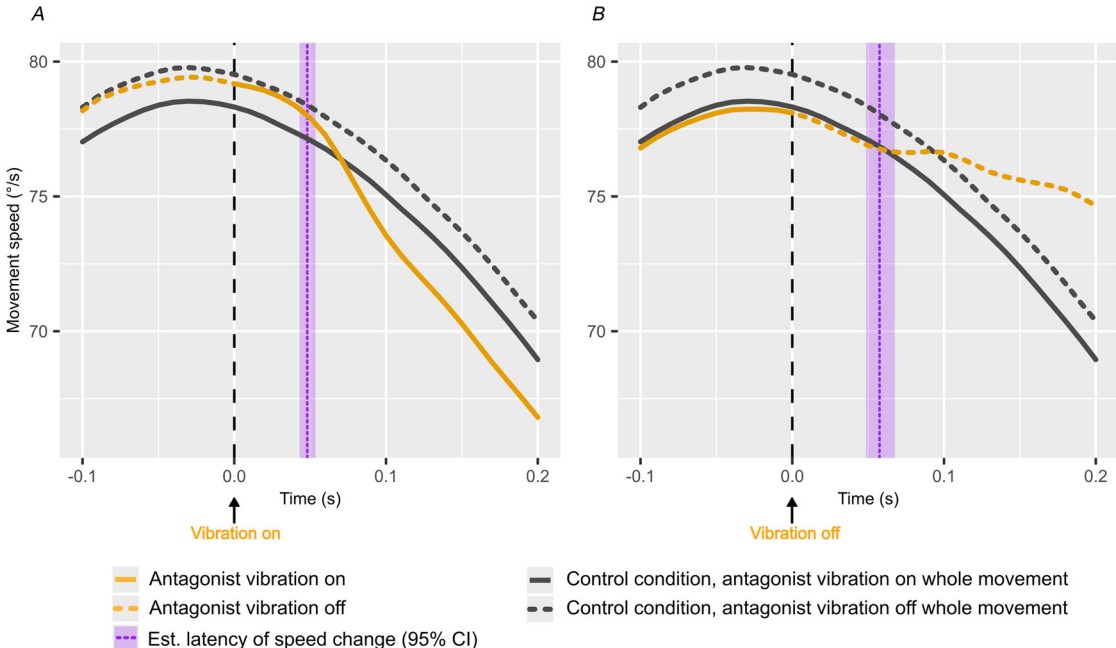

**Figure 6. Latency of movement speed control**
Panels *A* and *B* each show how movement speed is affected immediately after turning on (*A*) and off (*B*) the antagonist vibration motor 20° into an 80° extension movement. Overlaid is the maximum *a posteriori* (MAP) and 95% credible interval (CI) estimated group mean latency from vibration start/stop until movement speed is affected in purple. Grey lines are used to indicate the two control conditions, with the motor on or off throughout the movement. Orange lines show the intervention conditions, where the motor turns on (*A*) and off (*B*) 20° into the movement, time-locked to 0. Dotted line segments indicate when the motor is off, whereas solid lines indicate when the motor is on. Each line represents the mean movement speed of 816 trials, 48 per vibration condition per participant (*N* = 17).

fully drawn grey line. Figure 6*A* shows how turning on the antagonist vibration motor causes subjects to begin decreasing their movement speed at an estimated group mean latency of 48.2 ms (95% CI [42.9, 53.5], with a subject-to-subject SD of 1.0 ms (95% CI [0.2, 9.9]). Similarly turning off the antagonist vibration motor causes subjects to begin increasing their movement speed with a latency of 58.0 ms (95% CI [48.7, 67.7]), and with a subject-to-subject SD of 11.3 ms (95% CI [3.3, 22.8]). In both cases the change in movement speed is greater than the difference in movement speed between the two control conditions, where antagonist vibration was on or off for the entire movement.

## Discussion

We set out to investigate how sensed limb position and velocity are integrated into a cohesive perception of limb state and movement. Although the individual effects of muscle vibration on endpoint errors, changes in movement speed and the reportable perception of limb state have all been described in the existing literature, we sought here to provide a comprehensive picture of how these factors interact with one another during active movements.

Our investigation was based on the hypothesis that proprioceptive positional and velocity signals interact in a reciprocally reinforcing manner. Specifically we explored the prediction that velocity signals contribute to the perceived position of the arm during reaching movements.

In the following subsections we will discuss the findings from each of the three tasks in turn, addressing the three research questions outlined in the introduction.

### Muscle vibration biases both perceived and actual movement speed

In Task 1, our results show that antagonist muscle vibration causes participants to overestimate their movement speed while simultaneously decreasing their actual movement speed, whereas agonist muscle vibration increases movement speed. Although the perception of movement speed is only reported after the trial, it seems likely that the changes in movement speed arise as a corrective response to biased velocity feedback. This effect seems likely to be caused by the action of muscle vibration on the muscle stretch-velocity sensing capabilities of the muscle spindle afferents. Although both type Ia and type II afferents include velocity components in their signalling (Roll & Vedel, 1982) the primary perceptual effects of muscle vibration on the perception of movement are believed to be caused by type Ia afferent activity (Banks et al., 2021; Proske & Gandevia, 2012).

Our results indicated only clear perceptual effects of antagonist vibration on the perception of movement speed; if there is an effect of agonist vibration as well it is small enough not to show itself clearly with our experimental set-up and analysis approach. However, when concurrent vibration of both motors was applied, agonist vibration nevertheless nullified the effect of antagonist vibration, indicating that feedback from both the agonist and antagonist muscles contributes to the perception of limb state; similarly changes in movement speed were also cancelled out when both motors were active. This broadly matches previous findings in passive conditions, where vibration of flexor-extensor pairs has been found to cancel each other out (Gilhodes et al., 1986); when further considered together with findings indicating no effects of agonist vibration alone (Capaday & Cooke, 1981, 1983; Inglis & Frank, 1990), the general pattern indicates that vibration of the actively contracting and shortening muscle has smaller effects than the vibration of the passively lengthening antagonist. To the extent that there are effects, however, they appear to mirror what would be expected of vibrating the same muscle at rest. This is similarly seen in our results from Task 2, and matches the findings of Eschelmuller et al. (2025), who similarly showed smaller effects of agonist vibration compared to antagonist vibration on endpoint errors.

Although this might suggest that the CNS generally relies less on feedback from the agonist muscle during active movement, as has previously been suggested (Capaday & Cooke, 1981), it might also be an effect of a shift in what is signalled by the muscle spindle afferents. During active movement, muscle spindle activity is modulated by gamma-drive (Dimitriou, 2022), and even if agonist muscle vibration biases the firing frequency to a similar extent as is seen in antagonist vibration, the frequency shift may therefore not lead to an identical effect on inferred limb state.

By further considering our analysis of the subset of trials with no vibration and normal visual feedback we observe that variability in movement speed is generally predictive of the reported perception of movement speed. Specifically, trials performed with higher movement speeds are associated with underestimating the movement speed of the arm, and vice versa. This supports the notion that the effects of muscle vibration on both perceived and actual movement speed are not two concurrent but independent effects, both caused by muscle vibration. Instead, it appears that these two effects are indeed interdependent; it seems likely that as sensed movement speed becomes biased, a corrective speedup or slowdown is performed, perhaps to match an intended movement speed profile. Movement speed profiles of reaching movements are typically bell-shaped and smooth, a feature that has been well accounted for by models based on signal-dependent noise and Optimal Feedback Control

(Harris & Wolpert, 1998; Todorov & Jordan, 1998), as well as through models that minimise variational free energy (Friston et al., 2010). From either of these perspectives, any error in the perception of movement speed, whether arising spontaneously due to noise or through muscle vibration, would be expected to trigger the observed compensatory motor adjustments. These may align ongoing movement with either a planned optimal trajectory or with predictions produced by a generative model, depending on the assumed control framework.

It might further be considered if some of the changes in movement speed could be caused by a direct effect of muscle vibration biasing the sensed limb position; for example antagonist vibration could bias the sensed limb position through the effects on both types of muscle spindle afferents, such that the arm is sensed to be further along the movement than is the case, essentially adding a static offset to the perceived position. Thus sensing a position closer to the target than anticipated might similarly cause a reactive reduction in movement speed. However it is less clear how this effect would fit together with our findings of the associated changes in the perception of movement speed.

## The interrelationship of perceived movement speed and position

In Task 2, the results mirror those in Task 1 regarding the effects of muscle vibration on movement speed. Building upon the argumentation in the previous section, these effects may similarly be assumed to be in response to an induced bias in the perception of movement speed. The observed changes in movement speed are furthermore accompanied by congruent changes in endpoint error, indicating biases in the perceived arm position at the end of the movement. This is seen in that antagonist vibration causes both decreased movement speed and a bias towards undershooting the target, with effects in the opposite direction for agonist vibration. Thus considering Tasks 1 and 2 together, our results indicate that vibration-induced biases in the perception of movement speed are associated with congruent biases in the perception of limb position. This observation supports the hypothesis that estimation of limb position and velocity is an integrated process, as predicted by both Active Inference (Friston et al., 2010) and Optimal Feedback Control (Wolpert et al., 1995) frameworks.

It should, however, be noted that both type Ia and type II muscle spindle afferents also provide position-based signals, which are also likely to be affected to some degree as well by muscle vibration. As such it is difficult to ascertain to what extent the errors in perceived position revealed through the endpoint errors are a downstream effect of the integration of velocity signalling, and to what extent they are more direct effects of biased position sensors.

This would tend to support a common source of information contributing to the perceived velocity and position, although it is not conclusive evidence. Velocity information, which is clearly demonstrated to be associated with variability in movement speed in Task 1, both during vibration and in non-perturbed trials, thus seems like a likely candidate for such a common source of information. Although velocity-based feedback does not directly communicate position, it does, in principle, provide a robust signal regarding change in position over the course of a movement; if the start position is well represented, for example, due to visual feedback, then the integral of perceived velocity would provide an additional source of information regarding the travelled distance.

It has previously been demonstrated that vision of the hand position pre-movement increases the accuracy and precision of endpoint errors (Desmurget et al., 1997; Rossetti et al., 1994), and when the starting position of the hand is visually biased before movement start, an offset in the endpoint error is produced (Rossetti et al., 1995), which both would tend to support the usage of change-in-position information for position inference. Similarly over many sequential reaching movements, where proprioceptive position sense may drift substantially, individual movements remain well formed and are of very consistent direction and length (Brown et al., 2003a, 2003b; Patterson et al., 2017), which likewise points to a significant dependence on rate-of-change signals compared to absolute position signals. The findings by Cody et al. (1990), which demonstrated cumulative effects of vibration duration on endpoint errors, as well as correlated changes in movement speed and endpoint error dependent on vibration amplitude, further tend to support the idea of proprioceptive velocity information being used for endpoint position estimation, but are somewhat confounded by the fact that movement time was required to be constant across all conditions; similar effects have been found by Eschelmuller et al. (2023), who likewise demonstrated antagonist vibration-induced undershooting of a target across six different vibration conditions, with concurrent decreases in movement speed found in all conditions except the lowest frequency one at 20 Hz. Although keeping movement speed completely self-paced in the current set-up does increase the general variability in movement speed, it nevertheless allows for a more direct interpretation of the correlation between variability in endpoint errors and movement speed.

During active control of movement the CNS appears to rely on inferred movement velocity to trigger movement sequences at the correct timing to account for sensory and motor delays inherent to nervous conduction and processing (Cordo, 1988; Cordo et al., 1994). For example

opening the hand at the correct arm angle during an arm extension movement, as is required for accurate throwing, requires accounting for both efferent and afferent delays; if the command to open the hand is sent only once the afferent feedback from the arm communicates that the target angle has been reached, then the hand would be opened too late, by an amount corresponding, at least, to the sum of the afferent and efferent delays. Such observations are well in line with predictive frameworks of sensorimotor control, where internal generative models are thought to bridge the temporal gap between delayed sensory feedback and efferent motor output (Friston, 2011; Todorov & Jordan, 2002). These timing and coordination effects pose an additional challenge for interpreting how muscle vibration affects the currently inferred position and velocity, and how the inferred position is expected to change in the immediate future. For example it might well be imagined that even if an induced velocity offset does not affect the currently perceived position, it might still affect a forward prediction about the future position of the hand, prompting a different timing of pushing the button to indicate the position of the target. However, our analysis reveals small but consistent changes in movement time that work in the opposite direction. For example if antagonist vibration affected only the forward-predicted timing of reaching the target, we should expect to see decreased movement time; however to the extent that we do see any effects on movement time, they are in the opposite direction as to what would be expected, if the above described change in timing was a primary contributor to the changes in endpoint error that we see. Generally in a simplified sense, undershooting the target might have been associated with either moving slower for an unchanged movement time or moving for a shorter time at a similar speed. Instead we tend to see increased movement time in response to antagonist vibration, effectively increasing the required decrease in movement speed to achieve the same shift in endpoint error towards undershooting the target, and vice versa for agonist vibration. As it is, we are not able to provide any explanation for the observed changes in movement time, and further research is warranted to explore how these findings fit in the overall pattern of effects.

Across endpoint error, movement speed and movement time, we also analysed the effects of concurrent vibration of the agonist and antagonist motor together. Our results consistently reveal no statistical interaction between the two vibration motors across all three outcomes and both movement directions, indicating that the combined effect of both motors together is well approximated by the sum of the effect of each vibration motor alone. This means that the two motors, for the most part, tended to cancel each other out, leading to no overall effect when both motors were on; where effects were observed these were in the same direction as antagonist vibration, but of smaller size.

These results support the notion that feedback from both the agonist and antagonist is combined when inferring the overall state of the arm.

Finally we did not find evidence indicating that muscle vibration causes a decrease in endpoint error precision. Although our results match those reported by Eschelmuller et al. (2025), they are generally contrary to previous findings, which have indicated that muscle vibration may decrease the precision or weight placed on proprioceptive signals. This has been observed in tasks where antagonist-pair double vibration has been used to increase the strength of a visual illusion (Chancel & Ehrsson, 2023; Chancel et al., 2016), which the authors described in terms of decreased relative weighting of proprioceptive feedback due to degradation. Additionally it has been observed that such double muscle vibration leads to increased errors in angle matching and force production (Bock et al., 2007). Our results indicate no decrease in precision.

### Online control of movement speed

In Task 3 our results show how movement speed is affected immediately following turning on or off the antagonist vibration motor. The expected decrease in movement speed in response to antagonist vibration had an onset latency of approximately 48 ms. This finding is well in line with the findings by Keyser et al. (2019), who demonstrated a latency of changed force-production at 62 ms after onset of biceps vibration, and similarly matches EMG-based research, which has identified changes in muscle activity at latencies of 40–60 ms (Capaday & Cooke, 1983; Cody et al., 1990). Our observations further expand on this work, demonstrating that the previously observed latency of muscle force and EMG response matches well with actual changes in movement speed.

This confirms that the initial movement speed correction is a reflexive response to muscle vibration, likely mediated by increased firing of the antagonist type Ia afferents, operating well below the time required for conscious and deliberate motor control. In the context of Active Inference, such adjustments would reflect actions aimed at minimising proprioceptive prediction error, effectively aligning actual sensory feedback with predicted sensory consequences of movement (Friston et al., 2010).

We generally observed a similar, although slightly longer (approximately 59 ms) and more variable delay to turning off the antagonist vibration until movement speed was increased in response. This may be due to a limitation in our implementation of vibration motor control; although the vibration motor was always in a consistent state at vibration start (non-vibrating), the

state of the motor was variable when vibration stop was triggered, such that it might either be accelerating towards or away from the antagonist muscle when power was cut. Given that the length of a single oscillatory cycle exceeds the estimated difference (a single cycle lasts 12.5 ms for an 80 Hz signal), and that it likely takes some small amount of time for the moving motor to dissipate its energy from when power is cut, the actual physiological delay is likely very similar.

### Limitations and concluding remarks

In this study we approached the investigation of the inference of arm movement speed and position by applying muscle vibration during self-paced active movements, with arm movement restricted to the elbow joint only. This approach has the advantage of simplifying the assumptions regarding the effects of muscle vibration on perceived joint angles and velocities. However with both the biceps and triceps brachii crossing both the elbow and shoulder joint, any bias introduced in the firing rate of muscle afferents here would presumably influence both the perceived elbow and shoulder state. This significantly increases the complexity of forming hypotheses regarding the expected movement corrections in response to muscle vibration for whole-arm movement. Although the current approach with restricted movement sidesteps this issue to some extent it has drawbacks in terms of generalising our observations to more natural whole-arm/body movement. A limitation here is that one-dimensional movement, as is the case here, limits endpoint errors to lie along this dimension exclusively. Additional degrees of freedom may be necessary to fully describe whether muscle vibration increases variation in endpoint errors, even if this effect is not apparent with the current set-up.

Our results expand on earlier findings by directly demonstrating that both perceived and actual movement speed are affected in opposite directions by muscle vibration; we also show that the probability of over-estimating one's movement speed is increased for slower movements and vice versa. This was investigated during active movements, through the novel approach of requiring the participants to rate the relative speed of a virtual arm. This further provides a link to the inferred position of the hand during movement, through the congruent effects of muscle vibration on endpoint errors and movement speed, as well as the baseline correlation between these two outcomes. It should be emphasised here that velocity feedback is not argued to represent the only source of positional information during movement; there are, in general, many reasons for variability in movement speed which do not depend on mis-estimation of movement speed, and there are multiple sources of information which can lead to errors in inferred position which do not depend on proprioceptive velocity-based feedback. Additionally we show no increase in spread of endpoint errors in response to either agonist vibration, antagonist vibration or both together, which would be indicative of general degradation of downweighing of proprioceptive feedback, as has been previously suggested. In summary our results suggest that the perception of limb position and velocity is an integrated process, with sensed limb velocity likely contributing to the perceived limb position through an integral-like computation of the velocity signal. A mechanism for such velocity-to-position estimation arises naturally within predictive frameworks of sensorimotor control, such as Optimal Feedback Control and Active Inference (Friston et al., 2010; Wolpert et al., 1995). To the extent that velocity signals contribute to position estimation via an integration-like process it is equally plausible that position signals could influence velocity estimation through a differentiation-like mechanism, although this was not investigated here. Such reciprocal interactions in state estimation would serve to maintain a stable and self-consistent representation of limb state across both position and velocity domains, with the unavoidable downside that any sensory error in one domain would have downstream effects on the other domain as well, as demonstrated through the application of muscle vibration here. Although the observed movement speed corrections are too fast to be deliberately driven, the underlying perceived bias in movement speed is nevertheless available for verbal report.

To reach out and grasp an object we must infer our own body position and movement kinematics. This study provides further support for predictive models of sensory integration and motor control, demonstrating that limb state estimation is an integrated process spanning both position and velocity domains.

# Appendix

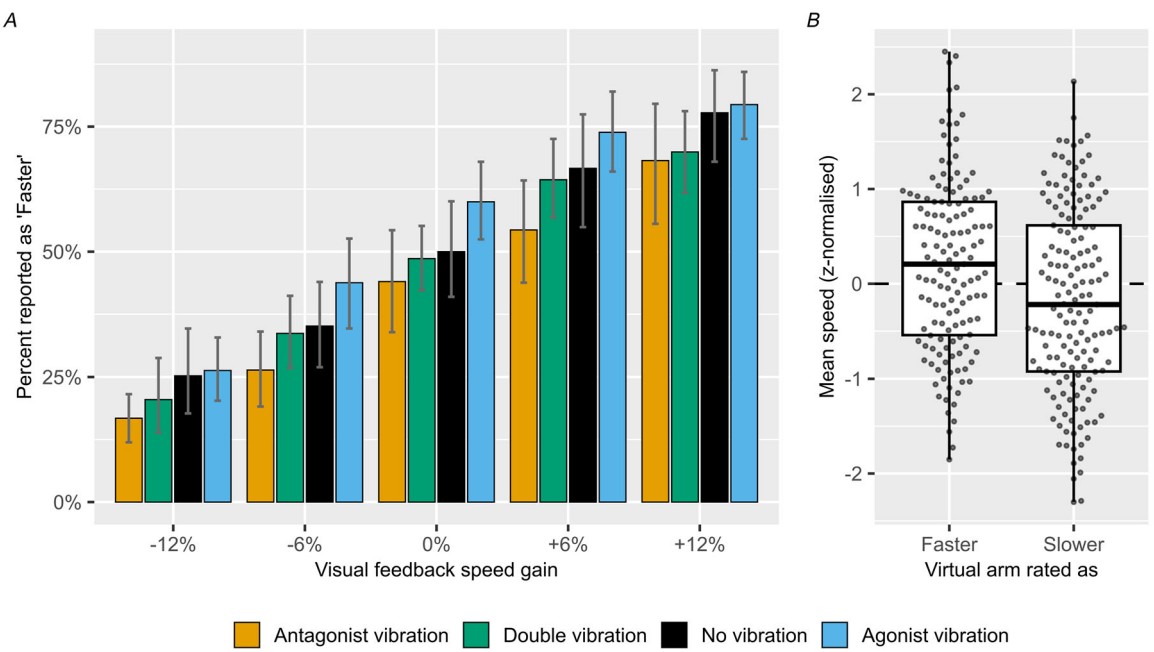

**Figure A1. Data distribution from Task 1**

Panel *A* shows the mean and 95% bootstrapped confidence intervals of subject percentages of trials reported as 'Faster' from Task 1, complement of Fig. 2*A*. Panel *B* shows z-normalised (within-subject) mean movement speed by subject rating of 'Faster' and 'Slower' from Task 1 trials with no visual speed gain (normal speed) and no vibration, complement of Fig. 5*B*.

**Table A1. Task 2 baseline performance.** Summary of mean (SD) endpoint errors, mean movement speed and movement time in Task 2 for the baseline trials with no muscle vibration, along with within- and between-subject variability. Based on 24 trials per movement direction per participant (*N* = 17).

| Outcome | Movement direction | Mean of subject means | Mean of subject SDs |
|---|---|---|---|
| Endpoint error (°) | Extension | −3.28 (5.07) | 3.65 (1.05) |
| Endpoint error (°) | Flexion | −1.12 (3.49) | 3.69 (0.96) |
| Mean movement speed (°/s) | Extension | 40.92 (9.83) | 7.73 (2.57) |
| Mean movement speed (°/s) | Flexion | 38.55 (7.6) | 6.59 (1.74) |
| Movement time (s) | Extension | 1.46 (0.34) | 0.29 (0.09) |
| Movement time (s) | Flexion | 1.61 (0.3) | 0.29 (0.1) |

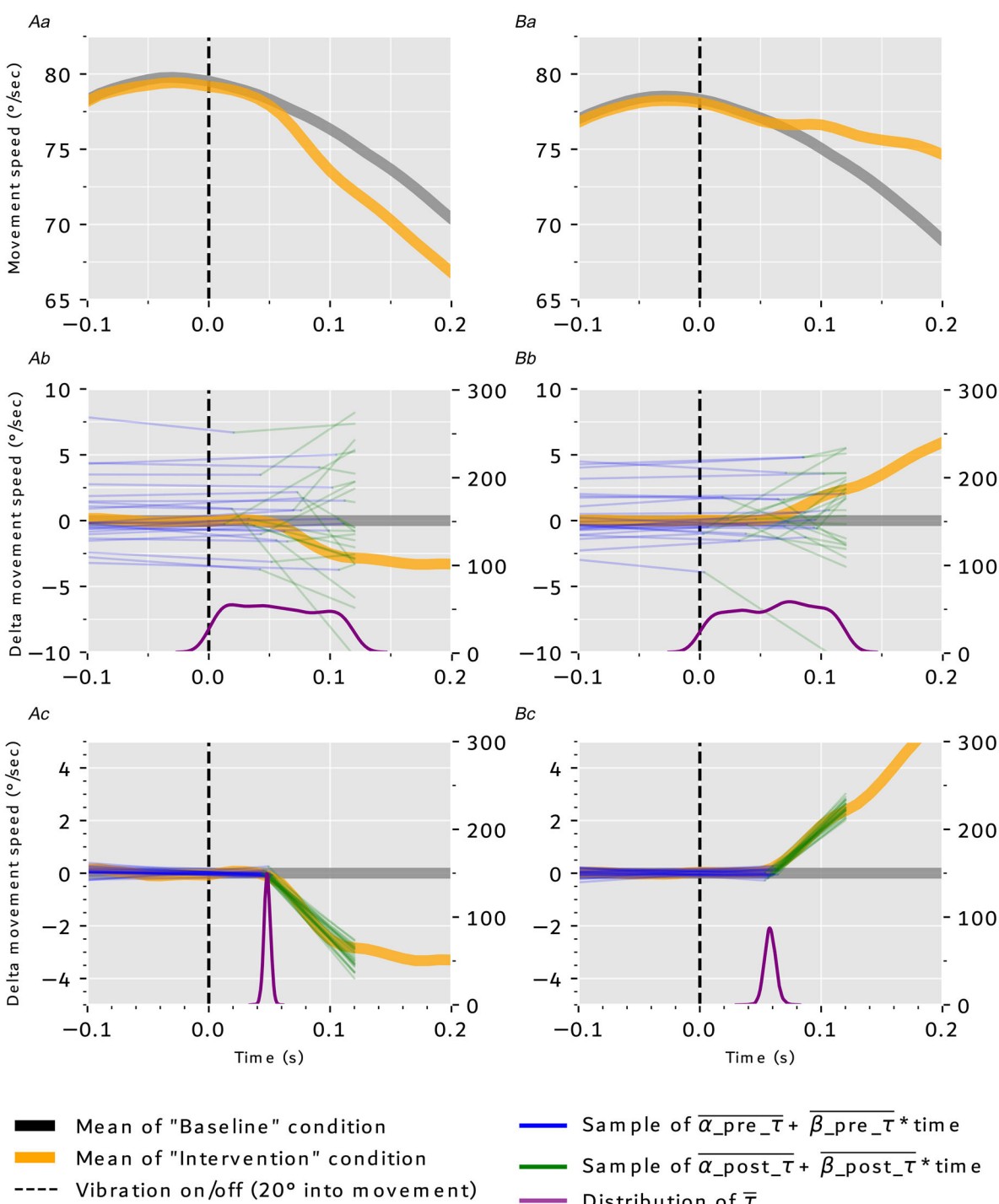

**Figure A2. Visualisation of the changepoint model fitting process**
Panels *Aa* and *Ba* show the mean movement speed of the two sets of intervention and control conditions, with control conditions in grey and intervention conditions in orange. Panels *Ab* and *Bb* display the same two lines, with the mean of the control condition subtracted from its corresponding intervention condition trials at each time point *t*. As a result the orange line now represents the delta mean movement speed between the two conditions. A subset of prior draws of the pre- and post-$\bar{\tau}$ linear function parameters (blue and green), as well as the prior distribution of $\bar{\tau}$ (purple), are also shown. Panels *Ac* and *Bc* show a subset of posterior draws of the two linear function parameters and the posterior distribution of $\bar{\tau}$. Note that although only the across-subjects mean delta movement speed is shown (orange), the model is fitted to the underlying trial-level data in a hierarchical fashion to account for subject-to-subject variation.

### Likelihood

$$\text{dms}_i \sim \mathcal{N}\left(\mu_i, \sigma_j\right)$$

$$\mu_i = \mu_{\text{pre\_}\tau, i} \cdot S_i + \mu_{\text{post\_}\tau, i} \cdot (1 - S_i)$$

$$S_i = \frac{1}{1 + e^{c_i}}$$

$$c_i = \frac{t_i - \tau_j}{0.001}$$

$$\mu_{\text{pre\_}\tau, i} = \alpha_{\text{pre\_}\tau, j} + \beta_{\text{pre\_}\tau, j} \cdot t_i$$

$$\mu_{\text{post\_}\tau, i} = \alpha_{\text{post\_}\tau, j} + \beta_{\text{post\_}\tau, j} \cdot t_i$$

### Priors

$$\alpha_{\text{pre\_}\tau, j} = \bar{\alpha}_{\text{pre\_}\tau} + \alpha_{\text{pre\_}\tau, \sigma} \cdot \varepsilon_{\alpha\_\text{pre\_}\tau, j} \qquad \text{for } j \in \{1, \ldots, n_{\text{id}}\}$$

$$\varepsilon_{\alpha\_\text{pre\_}\tau, j} \sim \mathcal{N}(0, 1) \qquad \text{for } j \in \{1, \ldots, n_{\text{id}}\}$$

$$\beta_{\text{pre\_}\tau, j} = \bar{\beta}_{\text{pre\_}\tau} + \beta_{\text{pre\_}\tau, \sigma} \cdot \varepsilon_{\beta\_\text{pre\_}\tau, j} \qquad \text{for } j \in \{1, \ldots, n_{\text{id}}\}$$

$$\varepsilon_{\beta\_\text{pre\_}\tau, j} \sim \mathcal{N}(0, 1) \qquad \text{for } j \in \{1, \ldots, n_{\text{id}}\}$$

$$\beta_{\text{post\_}\tau, j} = \bar{\beta}_{\text{post\_}\tau} + \beta_{\text{post\_}\tau, \sigma} \cdot \varepsilon_{\beta\_\text{post\_}\tau, j} \qquad \text{for } j \in \{1, \ldots, n_{\text{id}}\}$$

$$\varepsilon_{\beta\_\text{post\_}\tau, j} \sim \mathcal{N}(0, 1) \qquad \text{for } j \in \{1, \ldots, n_{\text{id}}\}$$

$$\alpha_{\text{post\_}\tau, j} = \alpha_{\text{pre\_}\tau, j} + \left(\beta_{\text{pre\_}\tau, j} - \beta_{\text{post\_}\tau, j} \cdot \tau_j\right) \quad \text{for } j \in \{1, \ldots, n_{\text{id}}\}$$

$$\tau_j = \bar{\tau} + \tau_\sigma \cdot \varepsilon_{\tau, j} \qquad \text{for } j \in \{1, \ldots, n_{\text{id}}\}$$

$$\varepsilon_{\tau, j} \sim \mathcal{N}(0, 1) \qquad \text{for } j \in \{1, \ldots, n_{\text{id}}\}$$

$$\sigma_j = \frac{1}{1 + e^{\sigma_{\log, j}}} \qquad \text{for } j \in \{1, \ldots, n_{\text{id}}\}$$

$$\sigma_{\log, j} = \bar{\sigma}_{\log} + \sigma_{\log, \sigma} \cdot \varepsilon_{\sigma\_\log, j} \qquad \text{for } j \in \{1, \ldots, n_{\text{id}}\}$$

$$\varepsilon_{\sigma\_\log, j} \sim \mathcal{N}(0, 1) \qquad \text{for } j \in \{1, \ldots, n_{\text{id}}\}$$

### Hyperpriors

$$\bar{\alpha}_{\text{pre\_}\tau} \sim \mathcal{N}(0, 2.5)$$

$$\alpha_{\text{pre\_}\tau, \sigma} \sim \mathcal{HN}(5)$$

$$\bar{\beta}_{\text{pre\_}\tau} \sim \mathcal{N}(0, 5)$$

$$\beta_{\text{pre\_}\tau, \sigma} \sim \mathcal{HN}(5)$$

$$\bar{\beta}_{\text{post\_}\tau} \sim \mathcal{N}(0, 50)$$

$$\beta_{\text{post\_}\tau, \sigma} \sim \mathcal{HN}(25)$$

$$\bar{\tau} \sim \mathcal{U}(0, t_{\max})$$

$$\tau_\sigma \sim \mathcal{HN}(0.025)$$

$$\bar{\sigma}_{\log} \sim \mathcal{N}(0, 10)$$

$$\sigma_{\log, \sigma} \sim \mathcal{HN}(10) \tag{A1}$$

The complete changepoint model, which was implemented in PyMC. See Fig. A2 for a visualisation of the fitting process.

The changepoint model approximates the delta mean movement speed as two linear functions, one before the effect of starting/stopping the vibration motor takes effect, with a slope of near 0, and one after, where the slope will indicate the estimated relative change in velocity over time, that is, acceleration. These assumptions allow us to sample the most likely changepoints ($\tau$) during model fitting. The same model, using identical priors, is fit twice, once for detecting the latency of turning on the vibration motor until velocity is changed, and once for turning off. $dms_i$ is the movement speed difference at time $t_i$ (in °/s). Specifically it is calculated by subtracting the mean speed of the control condition from each observation in the intervention condition (within-participant) at time $t_i$. A sigmoid function ($S_i$) with a very rapid transition between asymptotic values is used to blend between the two linear functions. The estimated inflection point $\tau$ thus models the latency. Zero-centred priors are used for the slopes both pre and post $\tau$, and a uniform prior was used for the value of $\tau$ between 0 (vibration start/stop) and the max time stamp of the included data (120 ms after vibration start/stop).

The model definition is subdivided into the likelihood function, as well as priors, which here describe the subject-level parameters (random effects, marked as $j$), and the hyperpriors, which represent the priors of the group-level parameters (fixed effects). The distribution of subject-level parameters is modelled as Gaussian distributions centred on the group means (marked with overlines); for example, subject changepoints are assumed to be normally distributed. We use an uncentred parameterisation of the model here to aid convergence. This relates to parameters marked as $\epsilon$; including these $\epsilon$ terms significantly improves sampling efficiency and convergence.

$dmi_i$ are at $t_i$ modelled as a Gaussian distribution with $\mu_i$ and $\sigma_j$. $\mu_i$ is a blend between the two linear functions, $\mu_{pre\_\tau,i}$, $\mu_{post\_\tau,i}$, blended using the value of a sigmoid function ($S_i$) at $t_i$. Of note $\alpha_{post\_\tau,j}$ is defined deterministically depending on the sampled values of $\alpha_{pre\_\tau,j}$, $\beta_{pre\_\tau,j}$, $\beta_{post\_\tau,j}$ and $\tau_j$, such that the two linear functions intersect at $t = \tau_j$. This ensures continuity between $\mu_{pre\_\tau,i}$ and $\mu_{post\_\tau,i}$ within each set of sampled values. For simplicity the distribution of observed $dmi_i$ is assumed to be constant across values of $t_i$, within-subject (denoted by $\sigma_j$).

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

## Additional information

### Data availability statement

The source data and analysis code are made publicly available at OSF:
- https://osf.io/869s4 (data)
- https://osf.io/7vdje (code).

### Competing interests

The authors report no conflicts of interest.

### Author contributions

The experiment was conducted at the Cognition, Intention and Action research group, Section for Cognition and Neuropsychology, Department of Psychology, University of Copenhagen, Øster Farimagsgade 2A, Copenhagen. E.S.M. and M.S.C. conceptualised and designed the research project. E.S.M. acquired and analysed the data and drafted the manuscript. E.S.M. and M.S.C. revised and edited the manuscript. E.S.M. and M.S.C. confirm that they have read and approved the final version of the manuscript.

### Funding

This project was supported by a DATA+ grant from UCPH. M.S.C. was supported by DFF-FKK (0132-00141B) and the Carlsberg Foundation (CF22-0941).

### Acknowledgements

The authors would like to thank Mantas Cibulskis, Difeng Yu and Joanna Emilia Bergström for their valuable discussions and insights on sensory inference.

### Keywords

inference, Ia afferent, motor control, movement speed, muscle spindle, muscle tendon vibration, perception, predictive coding, proprioception, position estimation, sensorimotor control, sensory integration, velocity estimation

### Supporting information

Additional supporting information can be found online in the Supporting Information section at the end of the HTML view of the article. Supporting information files available:

**Peer Review History**

