## [Peer Review History · The Journal of Physiology]

Proprioceptive integration in motor control

Erik Skjoldan Mortensen and Mark Schram Christensen

DOI: 10.1113/JP289835

Corresponding author(s): Erik Mortensen (esm@psy.ku.dk)

The following individual(s) involved in review of this submission have agreed to reveal their identity: Luc P. J. Selen (Referee #3)

Review Timeline:

Submission Date:	31-Jul-2025
Editorial Decision:	12-Sep-2025
Revision Received:	08-Dec-2025
Editorial Decision:	09-Feb-2026
Revision Received:	17-Feb-2026
Accepted:	18-Feb-2026

Senior Editor: Vaughan Macefield

Reviewing Editor: Jacob Thorstensen

Transaction Report:

Dear Dr Mortensen,

Re: JP-RP-2025-289835 "Proprioceptive integration in motor control" by Erik Skjoldan Mortensen and Mark Schram Christensen

Thank you for submitting your manuscript to The Journal of Physiology. It has been assessed by a Reviewing Editor and by 2 expert referees and we are pleased to tell you that it is potentially acceptable for publication following satisfactory major revision.

REVISION CHECKLIST:

Please upload two versions of your manuscript text: one with all relevant changes highlighted and one clean version with no

changes tracked. The manuscript file should include all tables and figure legends, but each figure/graph should be uploaded as separate, high-resolution files.

We look forward to receiving your revised submission.

Yours sincerely,

Vaughan Macefield
Senior Editor
The Journal of Physiology

REQUIRED ITEMS FOR REVISION

- Author photo and profile. First or joint first authors are asked to provide a short biography (no more than 100 words for one author or 150 words in total for joint first authors) and a portrait photograph. These should be uploaded and clearly labelled together in a Word document with the revised version of the manuscript. See Information for Authors for further details.
- You must start the Methods section with a paragraph headed Ethical Approval. If experiments were conducted on humans, confirmation that informed consent was obtained, preferably in writing, that the studies conformed to the standards set by the latest revision of the Declaration of Helsinki and that the procedures were approved by a properly constituted ethics committee, which should be named, must be included in the article file. If the research study was registered (clause 35 of the Declaration of Helsinki), the registration database should be indicated, otherwise the lack of registration should be noted as an exception (e.g. The study conformed to the standards set by the Declaration of Helsinki, except for registration in a database). For further information see: <https://physoc.onlinelibrary.wiley.com/hub/human-experiments>.
- Please upload separate high-quality figure files via the submission form.
- Please ensure that the Article File you upload is a Word file.
- A Data Availability Statement is required for all papers reporting original data. This must be in the Additional Information section of the manuscript itself. It must have the paragraph heading 'Data Availability Statement'. All data supporting the results in the paper must be either: in the paper itself; uploaded as Supporting Information for Online Publication; or archived in an appropriate public repository. The statement needs to describe the availability or the absence of shared data. Authors must include in their statement: a link to the repository they have used, or a statement that it is available as Supporting Information; reference the data in the appropriate sections(s) of their manuscript; and cite the data they have shared in the References section. Whenever possible, the scripts and other artefacts used to generate the analyses presented in the paper should also be publicly archived. If sharing data compromises ethical standards or legal requirements then authors are not expected to share it, but must note this in their statement. For more information, see our Statistics Policy.
- Please include an Abstract Figure file, as well as the Figure Legend text within the main article file. The Abstract Figure is a piece of artwork designed to give readers an immediate understanding of the research and should summarise the main conclusions. If possible, the image should be easily 'readable' from left to right or top to bottom. It should show the physiological relevance of the manuscript so readers can assess the importance and content of its findings. Abstract Figures should not merely recapitulate other figures in the manuscript. Please try to keep the diagram as simple as possible and without superfluous information that may distract from the main conclusion(s). Abstract Figures must be provided by authors no later than the revised manuscript stage and should be uploaded as a separate file during online submission labelled as File Type 'Abstract Figure'. Please also ensure that you include the figure legend in the main article file. All Abstract Figures

should be created using BioRender. Authors should use The Journal's premium BioRender account to export high-resolution images. Details on how to use and access the premium account are included as part of this email.

- Please include a full title page as part of your main article (Word) file, which should contain the following: title, authors, affiliations, corresponding author name and contact details, keywords, and running title.

- Please ensure that all figures and tables have a title and legend, and that they have been cited within the main article text.

EDITOR COMMENTS

Reviewing Editor:

This is an interesting submission unpacking mechanisms of proprioceptive integration and its effects on voluntary movement. It has been reviewed by two experts, who have praised some aspects of the submission, but have also identified many shortcomings. Specifically, there is a lack of methodological details, particularly regarding some statistics and modelling, and some claims/conclusions raised by authors are not entirely substantiated. Although the results are useful for those interested in sensorimotor control, they are perhaps not entirely surprising and don't represent a major significant advance in this area. Nonetheless, the referees have provided many valuable comments that I hope the authors deem useful for their paper moving forward.

Senior Editor:

Thank you for submitting your manuscript to the Journal of Physiology. I have now received comments from two independent reviewers and the reviewing editor, all experts in the field. As you will see from their comments, the reviewers acknowledge that there are some interesting elements in your manuscript yet many missing methodological details, which I now invite you to address. You will need to fully revise the manuscript according to the reviewers' comments, and you will need to convince me that your study makes a significant difference advance to the field; as it is written, myself and the reviewing editor are not convinced. You will also need to clarify that vibration will indeed activate group II afferents, and - if the muscle is not fully relaxed - also GTOs.

REFEREE COMMENTS

Referee #2:

This study addresses an important question in proprioceptive integration, and the experiment overall seems to be well done and of interest to the current body of literature. There are some major issues that need to be addressed to allow for full interpretation of the experiments and relation back to the body of literature. The main issues are:

1. Incomplete description of the proprioceptive literature (especially Ia vs II afferents)
2. Insufficient methodological detail
3. Lack of clarity and justification in the statistical modeling
4. Incomplete presentation of results (notably double vibration)
5. Over-interpretation of correlations as evidence for integration of velocity into position.

Addressing these points would substantially improve the rigor, accessibility, and impact of the paper.

Introduction

- Please operationalize your definition of proprioception in the beginning of the introduction. While the term is used widely, in my experience it is used with relatively widely varying definitions.

- Paragraph 3: "...for these reflexes are the type Ia and type II afferent muscle spindles". This seems to be implying that there are two different types of muscle spindles, instead of two different types of afferents innervated a given muscle

spindle.

- Paragraph 3: It may be relevant here to at least discuss more recent data and suggestions of muscle spindles acting as dynamic signal processing devices (Dimitriou, 2022) or as load sensors (Blum et al., 2017, 2020).

- Paragraph 4: It is important to note that group II afferents will also respond to tendon vibration, albeit not as strongly. However, it is slightly misleading to only mention the Ia responses.

- Paragraph 5: The work of Cordo and colleagues here would be quite relevant (for example see P. Cordo, Bevan, et al., 1995; P. Cordo, Gurfinkel, et al., 1995; P. J. Cordo, 1988).

- Paragraph 5: Vibration applied to the agonist has historically shown no effect, however, a recent study has found a bias towards overshooting in an elbow targeting task. (Eschelmuller et al., 2025), and may be of interest to the authors here.

- Paragraph 8: The conclusions here need to be discussed more concretely as there is missing discussion of important points. The authors claim that the changes in perceived position are due to integration of velocity signals from the Ia afferents. It is critical to note two points here: Ia afferents can provide a robust length code as well as velocity, particularly when there is a strong gamma-static motoneuron activation. Therefore, the perceived position does not need to be integrated from a velocity estimate as the positional information is present. Secondly, these statements are based on tendon vibration studies, which the authors have stated only activate the primary Ia afferents, however, this is not true, and group II afferents can also be biased by tendon vibration, particularly at lower frequency ranges. For example, depending on what the participant was asked to focus on, tendon vibration could produce changes in the perceived position or the perceived velocity (Sittig et al., 1985). Additionally, vibration of 20 Hz during a wrist extension task, there was evidence for undershooting with no changes in velocity (Eschelmuller et al., 2023). The authors need to include a more nuanced discussion here or be more concrete in the assumptions they are making as this is a critical component of the goal of this project.

- The methods make use of double vibration, however, there is no mention of the current knowledge of the effects of agonist and antagonist vibration together in the introduction. See for example Bellan et al., 2016; Bock et al., 2007; Bock & Thomas, 2011; Eschelmuller et al., 2025; Fuentes et al., 2012; Gilhodes et al., 1986; Longo et al., 2009.

Methods:

- What was the monetary value that the participants were compensated?

Task 1:

- What speed were the participants asked to move at? Were they trained at this speed during the practice?

Task 2: Please include more detail of the task instruction for the participant.

- What is meant by indicate the location of the target by moving their arm and pressing a thumb button.

- Were they aligning their thumb, index finger etc.?

- Were they allowed to make corrections?

- How fast were they told to move?

Task 3: Please also include more details

- How fast were they told to move?

- Please explain/justify why you chose to make the comparisons you have focused on. Why 0-20 and 20-80 degrees.

Rotating armrest:

- More detail is needed on the collection of elbow angle/hand position. Was it only the VR controller? What is the sampling frequency of this controller? Spatial resolution?

Vibration motors:

- If you wanted to avoid changes in pressure, why wasn't the triceps vibrator also secured onto the muscle belly?

- What was the amplitude of the vibration?

- The make/model is present but also explain what type of vibrator was used. For example, was it a linear motor, eccentrically mounted DC motor etc. Each will have different characteristics.

Data processing and statistical modeling

- Overall, the modeling needs to be more clearly justified and explained for the reader. Why were Bayesian models chosen, and how do they directly inform the study's hypotheses? Why was a Bayesian as opposed to a frequentist approach used? As written, a reader who is not already familiar with the brms ecosystem will struggle to follow the modeling section. The main manuscript text should describe the models in clear, well-defined, and broadly accepted statistical terms, not in package-specific code syntax. For example, instead of presenting code snippets, the authors could provide the underlying model equations (e.g., linking predictors and outcomes in regression form) or a schematic/diagram that illustrates the flow of the model. Since the exact code is published alongside the manuscript, the main text should focus on conceptual clarity: what question each model addresses, why it is structured that way, and how it connects to the theoretical framework.

- Overall, the models themselves seem technically correct to my knowledge, but this section needs to be significantly reworked with the goal of allowing the reader to understand the rationale and purposes of each model.

Results:

- Overall, the results were not fully presented and explained. Many details have been omitted, and some analyses are never mentioned even though they are present in some of the figures.

- Task 1 results. There is no discussion or visualization of the effects of the double vibration results here. Additionally, it is unclear why only the visual speed gain of 6% is visualized in 2B and 2C. In figure 2B, the reference is no vibration. Presumably this was from the visual speed gain of 100%. Similarly, were the agonist and antagonist vibration errors also calculated based on the 100% relative movement speed. Potentially this information could be inferred from the modelling section above, but as mentioned earlier this section is not very clear and it should be explicitly stated here.

- Task 2 results. This section is difficult to interpret as many details are missing from the task instruction and parameters in the methods. In the first line, figure 6 is reference, however this seems incorrect. As with task 1, the result of the double vibration is not sufficiently reported.

- Task 1 and 2 integrative results. Here the authors again state that position estimates are derived by integrating sensed movement speed. However, if this were the case, the correlations in figure 4 would be expected to be much stronger. If the end point errors are being driven by changes in movements speed, we should see robust correlations between the two measures. Tendon vibration likely drives changes in both perceived position of the hand and perceived velocity. The authors argument that both are derived by velocity is not grounded in previous work or their current data. The authors need to outline this idea in more detail (as mentioned above in the introduction), but also here based on the data that they have, as currently it is not clear.

Discussion

- Due to issues with data presentation and reporting of task parameters, I will only provide a high-level review of the discussion as without the extra details it is difficult to get a whole picture of the experiments. As with the discussion, the effects of dual vibration are essentially left out completely.

- 5.1 paragraph 3. The authors need to expand their viewpoint that vibration only biases the "velocity sensors". They bias both Ia and group II afferents, both of which can signal velocity and position. The relative amount of signal from each depends on the gamma motoneuron bias present.

- Overall section 5.1 doesn't provide a very detailed discussion of the data they collected and how this links back to their original hypotheses and the models they mention (OFC and AI).

- Section 5.2 requires the data to demonstrate that changes in movement speed cause a change in end point position. In my opinion the authors have not clearly shown this cause-and-effect relationship. It could be that vibration biased both perceived position and movement, and both are used to determine end point. The correlations that have been presented in figures 4 and 5 are not convincing enough to make these statements. The discussion of the paradoxical effects on movement time is not clear, and the authors should provide a more detailed explanation on how these fits with their model of the task and results.

References

Bellan, V., Wallwork, S. B., Stanton, T. R., Reverberi, C., Gallace, A., & Moseley, G. L. (2016). No Telescoping Effect with Dual Tendon Vibration. *PLOS ONE*, 11(6), e0157351. <https://doi.org/10.1371/journal.pone.0157351>

- Blum, K. P., Campbell, K. S., Horslen, B. C., Nardelli, P., Housley, S. N., Cope, T. C., & Ting, L. H. (2020). Diverse and complex muscle spindle afferent firing properties emerge from multiscale muscle mechanics. *eLife*, 9, e55177. <https://doi.org/10.7554/eLife.55177>
- Blum, K. P., D'Incamps, B. L., Zytnicki, D., & Ting, L. H. (2017). Force encoding in muscle spindles during stretch of passive muscle. *PLOS Computational Biology*, 13(9), e1005767. <https://doi.org/10.1371/journal.pcbi.1005767>
- Bock, O., Pipereit, K., & Mierau, A. (2007). A method to reversibly degrade proprioceptive feedback in research on human motor control. *Journal of Neuroscience Methods*, 160(2), 246-250. <https://doi.org/10.1016/j.jneumeth.2006.09.010>
- Bock, O., & Thomas, M. (2011). Proprioception plays a different role for sensorimotor adaptation to different distortions. *Human Movement Science*, 30(3), 415-423. <https://doi.org/10.1016/j.humov.2010.10.007>
- Cordo, P., Bevan, L., Gurfinkel, V., Carlton, L., Carlton, M., & Kerr, G. (1995). Proprioceptive coordination of discrete movement sequences: Mechanism and generality. *Canadian Journal of Physiology and Pharmacology*, 73(2), 305-315. <https://doi.org/10.1139/y95-041>
- Cordo, P., Gurfinkel, V. S., Bevan, L., & Kerr, G. K. (1995). Proprioceptive consequences of tendon vibration during movement. *Journal of Neurophysiology*, 74(4), 1675-1688. <https://doi.org/10.1152/jn.1995.74.4.1675>
- Cordo, P. J. (1988). Kinesthetic coordination of a movement sequence in humans. *Neuroscience Letters*, 92(1), 40-45. [https://doi.org/10.1016/0304-3940\(88\)90739-2](https://doi.org/10.1016/0304-3940(88)90739-2)
- Dimitriou, M. (2022). Human muscle spindles are wired to function as controllable signal-processing devices. *eLife*, 11, e78091. <https://doi.org/10.7554/eLife.78091>
- Eschelmuller, G., Inglis, J. T., Kim, H., & Chua, R. (2025). Dual agonist and antagonist muscle vibration produces a bias in end point with no change in variability (p. 2025.03.03.641240). *bioRxiv*. <https://doi.org/10.1101/2025.03.03.641240>
- Eschelmuller, G., Szarka, A., Gandossi, B., Inglis, J. T., & Chua, R. (2023). The effects of periodic and noisy tendon vibration on a kinesthetic targeting task. *Experimental Brain Research*. <https://doi.org/10.1007/s00221-023-06727-1>
- Fuentes, C. T., Gomi, H., & Haggard, P. (2012). Temporal features of human tendon vibration illusions. *European Journal of Neuroscience*, 36(12), 3709-3717. <https://doi.org/10.1111/ejn.12004>
- Gilhodes, J. C., Roll, J. P., & Tardy-Gervet, M. F. (1986). Perceptual and motor effects of agonist-antagonist muscle vibration in man. *Experimental Brain Research*, 61(2). <https://doi.org/10.1007/BF00239528>
- Longo, M. R., Kammers, M. P. M., Gomi, H., Tsakiris, M., & Haggard, P. (2009). Contraction of body representation induced by proprioceptive conflict. *Current Biology*, 19(17), R727-R728. <https://doi.org/10.1016/j.cub.2009.07.024>

Sittig, A. C., Van Der Gon, J. J. D., & Gielen, C. C. A. M. (1985). Separate control of arm position and velocity demonstrated by vibration of muscle tendon in man. *Experimental Brain Research*, 60(3), 445-453. <https://doi.org/10.1007/BF00236930>

Referee #3:

Proprioceptive integration in motor control

Authors: Erik Mortensen and Mark Christensen

I read this paper with great interest. I believe it presents a coherent set of well-designed experiments on the role of afferent feedback on control and estimation, using muscle vibration. Also, the statistical methodology is powerful, using MCMC-techniques to unveil group level effects. This methodology is very well suited to analyse the highly variable responses to muscle vibration. The description of these methods well done and provides enough detail to reproduce. So, overall a great paper that I would recommend publishing.

Here some minor remarks:

[General experimental procedures] What is the precision of the alignment between the actual arm and the virtual arm and how was this established?

Task 1: Smart way of investigating the influence of vibration on movement speed.

Task 2: Manipulation to investigate vibration effects on actual movement speed changes.

Task 3: Time/position dependent changes in movement speed.

I like the sophisticated statistics with partial pooling in BRMS and the centered approach.

Figure 2: I would prefer visual speed gain as -12%, -6%, 6% and 12%, so the change, as this is how you report in the main text.

Part 4.2 - do you really intend to refer to Figure 6 here? Then maybe move the figure, you now make a very far fast forward.

Part 5.3 - You compare the behavioural onset latency wrt vibration onset, to EMG based measures. However, Keyser et al. (2019) have also reported behavioural latencies, so good to mention that you find them in the same range.

"From a signal-dependent noise perspective ..." - this sentence doesn't make sense. The mentioned framework doesn't include corrections. OFC is a signal-dependent noise feedback control framework and that explicitly does not restore the original trajectory. If you want to make a case for trajectory control in OFC, you should have a look at: Cluff, T., & Scott, S.

H. (2015). Apparent and actual trajectory control depend on the behavioral context in upper limb motor tasks. *Journal of Neuroscience*, 35(36), 12465-12476.

END OF COMMENTS

Response to referees

Our responses are marked with yellow text highlighting, while example text bites from the revised manuscript are highlighted with green.

Reviewing Editor:

This is an interesting submission unpacking mechanisms of proprioceptive integration and its effects on voluntary movement. It has been reviewed by two experts, who have praised some aspects of the submission, but have also identified many shortcomings. Specifically, there is a lack of methodological details, particularly regarding some statistics and modelling, and some claims/conclusions raised by authors are not entirely substantiated. Although the results are useful for those interested in sensorimotor control, they are perhaps not entirely surprising and don't represent a major significant advance in this area. Nonetheless, the referees have provided many valuable comments that I hope the authors deem useful for their paper moving forward.

- (This is a shared response to both editors)

We greatly appreciate the provided feedback, and we have now significantly reworked our submission, taking the feedback into account.

In particular, we substantially expanded the Methods and Results sections to provide a more thorough description of the experimental setup and the specification of statistical models, as well as the rationale for the choices of specific models.

Furthermore, we have expanded both our Introduction and the discussion and interpretation of results to provide a more nuanced interpretation, taking into account the position- and velocity-dependent signals provided by both type Ia and type II afferents, as well as the likely effect of muscle vibration on both of these receptor types, as well as Golgi tendon organ afferents.

We hope you agree that the presented data and results now warrant our interpretation as now formulated.

While our findings here are not paradigm-shifting, we nevertheless believe that they do contribute in a significant way to the understanding of the perception and control of movement speed, and how velocity information may be utilised to augment positional inference.

Senior Editor:

Thank you for submitting your manuscript to the Journal of Physiology. I have now received comments from two independent reviewers and the reviewing editor, all experts in the field. As you will see from their comments, the reviewers acknowledge that there are some interesting elements in your manuscript yet many missing methodological details, which I now invite you to address. You will need to fully revise the manuscript according to the reviewers' comments, and you will need to convince me that your study makes a significant difference advance to the field; as it is written, myself and the reviewing editor are not convinced. You will also need to clarify that vibration will indeed activate group II afferents, and - if the muscle is not fully relaxed - also GTOs.

- (This is a shared response to both editors)

We greatly appreciate the provided feedback, and we have now significantly reworked our submission, taking the feedback into account.

In particular, we substantially expanded the Methods and Results sections to provide a more thorough description of the experimental setup and the specification of statistical models, as well as the rationale for the choices of specific models.

Furthermore, we have expanded both our Introduction and the discussion and interpretation of results to provide a more nuanced interpretation, taking into account the position- and velocity-dependent signals provided by both type Ia and type II afferents, as well as the likely effect of muscle vibration on both of these receptor types, as well as Golgi tendon organ afferents.

We hope you agree that the presented data and results now warrant our interpretation as now formulated.

While our findings here are not paradigm-shifting, we nevertheless believe that they do contribute in a significant way to the understanding of the perception and control of movement speed, and how velocity information may be utilised to augment positional inference.

Referee #2:

This study addresses an important question in proprioceptive integration, and the experiment overall seems to be well done and of interest to the current body of literature. There are some major issues that need to be addressed to allow for full interpretation of the experiments and relation back to the body of literature. The main issues are:

1. Incomplete description of the proprioceptive literature (especially Ia vs II afferents)
2. Insufficient methodological detail
3. Lack of clarity and justification in the statistical modeling
4. Incomplete presentation of results (notably double vibration)
5. Over-interpretation of correlations as evidence for integration of velocity into position.

Addressing these points would substantially improve the rigor, accessibility, and impact of the paper.

- We greatly appreciate the detailed feedback, and we have now significantly reworked the submission, taking the comments into account. We believe our revised submission now represents a more thorough and nuanced description and discussion of our findings.
We address each of the points below with text references to how we have approached a resolution to the highlighted issue/shortcoming.
- As a brief overview, we have substantially expanded on the Introduction in order to cover the highlighted shortcomings, and we have incorporated much of the recommended literature, which now provides much improved nuance and coverage of the existing knowledge base.
The Methods section has undergone a major revision, including both the description of the experimental setup and the completed tasks, as well as the description and rationale of the chosen statistical models.
The results section has been expanded to better walk readers through the analysis results, and to also show our complete findings regarding double vibration. Figure 2 (main results of Task 1) has been reworked in accordance with this.
Our discussion and interpretation of the results is now substantially more nuanced, taking the critique of over-interpretation into account. We hope that you agree that our interpretation, as now formulated, is warranted given the results and analysis.

Introduction

- Please operationalize your definition of proprioception in the beginning of the introduction. While the term is used widely, in my experience it is used with relatively widely varying definitions.

- We added a paragraph in the introduction to clearly define what we here understand by proprioception and the scope of our discussion of it.

- In the introduction section: (line 63)

- “Proprioception describes the sensation of the position and movement of the body, which arises from a collection of peripheral mechanoreceptors found in the muscles, skin, and joints all around the body (Tuthill & Azim, 2018). Beyond giving rise to the perception of the state of the body, these feedback signals are also tightly integrated with motor control, as seen in cases of peripheral deafferentation (Cole & Waterman, 1995). To limit the scope of our introduction to the subject, we will here mainly consider the perceptual and behavioural effects of muscle spindle afferents, which are stretch-sensitive receptors found throughout the skeletal muscle system, where they innervate intrafusal muscle fibres (Proske & Gandevia, 2012).”

- Paragraph 3: "...for these reflexes are the type Ia and type II afferent muscle spindles". This seems to be implying that there are two different types of muscle spindles, instead of two different types of afferents innervated a given muscle spindle.

- We changed the wording the emphasise this point.

- In the introduction section: (line 79)

- “Two of the primary peripheral sensory organs responsible for these reflexes are the type Ia and type II muscle spindle afferents (Proske & Gandevia, 2012).”

- Paragraph 3: It may be relevant here to at least discuss more recent data and suggestions of muscle spindles acting as dynamic signal processing devices (Dimitriou, 2022) or as load sensors (Blum et al., 2017, 2020).

- We added a paragraph to highlight these more recent findings.

- In the introduction section: (line 162)

- “It should be emphasised that there is, in practice, significant functional overlap between the type Ia and type II afferents, such that they both encode position- and velocity-dependent signals to some extent, but with different primary functions. Their encoding of movement further depends on the activity of the parent muscle and any external load against which they are working; for example, acting against gravity may phase shift them towards signalling acceleration and velocity, respectively (Dimitriou & Edin, 2008; Macefield & Knellwolf, 2018; Banks et al., 2021). More recent work has indicated the possibility of muscle spindles functioning as peripheral processing units during active movements (Dimitriou, 2022), and that their signalling may be more closely correlated with muscle force, and its time derivatives (Blum et al., 2017, 2020). Together, such findings may indicate a split in function between the passively

lengthening antagonist muscle spindle afferents, whose activity appears to signal muscle length and stretch velocity and/or force (and its time-derivatives), while the afferents found in the actively contracting agonist muscle may be involved in processing that depends on the descending drive which activates the intrafusal fibres, facilitating motor control.”

- Paragraph 4: It is important to note that group II afferents will also respond to tendon vibration, albeit not as strongly. However, it is slightly misleading to only mention the Ia responses.

- We revised the phrasing to include this important nuance.

- In the introduction section: (line 84)

- “In addition to being sensitive to lengthening of the receptor-bearing muscle, both type Ia and type II afferents also respond to vibration of the muscle/tendon by increasing their firing rate up to the vibration frequency or a subharmonic of it (Roll *et al.*, 1989). This has prompted a large body of experimental research investigating the contents and inference of these proprioceptive signals. At a behavioural level, it has been demonstrated that the vibration of a muscle can bias both the perceived limb position and limb velocity (Goodwin *et al.*, 1972; Eklund, 1972; Roll & Vedel, 1982; Gilhodes *et al.*, 1986; Inglis *et al.*, 1991). It should further be noted that the Golgi tendon organ, in addition to both types of muscle afferents, is also affected by vibration (Roll *et al.*, 1989), making it difficult to disentangle the perceptual effects originating from each receptor type independently. The type Ia does, however, respond particularly well, and it is generally believed that the primary perceptual effects of muscle vibration originate from this receptor type (Proske & Gandevia, 2012).”

- Paragraph 5: The work of Cordo and colleagues here would be quite relevant (for example see P. Cordo, Bevan, et al., 1995; P. Cordo, Gurfinkel, et al., 1995; P. J. Cordo, 1988).

- We did indeed find these studies very relevant and have incorporated them in the discussion of Task 2 results.

- In the discussion: (line 947)

- “During active control of movement, the CNS appears to rely on inferred movement velocity to trigger movement sequences at the correct timing to account for sensory and motor delays inherent to nervous conduction and processing (Cordo, 1988; Cordo et al., 1994). For example, opening the hand at the correct arm angle during an arm extension movement, as is required for accurate throwing, requires accounting for both efferent and afferent delays; if the command to open the hand is sent only once the afferent feedback from the arm communicates that the target angle has been reached, then the hand would be opened too late, by an amount corresponding, at least, to the sum of the afferent and efferent delays.”

- Paragraph 5: Vibration applied to the agonist has historically shown no effect, however, a recent study has found a bias towards overshooting in an elbow targeting task. (Eschelmuller et al., 2025), and may be of interest to the authors here.

- This preprint proved very interesting to include as their findings match some of our own observations regarding agonist and dual vibration. In the revised manuscript we have made reference to this study on several occasions:
- In the introduction: (line 130)
- However, a recent study reported that agonist vibration can lead to a small overshoot effect, though it remains weaker than the undershoot caused by antagonist vibration in the same task (Eschelmuller et al., 2025). As in the passive condition, concurrent agonist-antagonist vibration has been found to have variable effects. Eschelmuller et al. (2025) demonstrated that concurrent vibration produces an effect approximately equal to the sum of their individual effects, with no increase in variable error, congruent with the notion of limb motion being inferred from the combined signalling from all muscles acting on the relevant joint.”
- We additionally refer to these effects in the discussion: (line 859)
- “... , the general pattern indicates that vibration of the actively contracting and shortening muscle has smaller effects than the vibration of the passively lengthening antagonist. To the extent that there are effects, however, they appear to mirror what would be expected of vibrating the same muscle at rest. This is similarly seen in our results from Task 2, and matches the findings of Eschelmuller et al. (2025), who similarly showed smaller effects of agonist vibration compared to antagonist vibration on endpoint errors.“

- Paragraph 8: The conclusions here need to be discussed more concretely as there is missing discussion of important points. The authors claim that the changes in perceived position are due to integration of velocity signals from the Ia afferents. It is critical to note two points here: Ia afferents can provide a robust length code as well as velocity, particularly when there is a strong gamma-static motoneuron activation. Therefore, the perceived position does not need to be integrated from a velocity estimate as the positional information is present. Secondly, these statements are based on tendon vibration studies, which the authors have stated only activate the primary Ia afferents, however, this is not true, and group II afferents can also be biased by tendon vibration, particularly at lower frequency ranges. For example, depending on what the participant was asked to focus on, tendon vibration could produce changes in the perceived position or the perceived velocity (Sittig et al., 1985). Additionally, vibration of 20 Hz during a wrist extension task, there was evidence for undershooting with no changes in velocity (Eschelmuller et al., 2023). The authors need to include a more nuanced discussion here or be more concrete in the assumptions they are making as this is a critical component of the goal of this project.

- We appreciate the feedback and have reworded the paragraph substantially to better reflect the nuances in type Ia/II signalling and how they may each contribute to the

perceived state of the limb. We found both of the referenced articles very relevant, and they have been incorporated in the introduction and discussion.

- The paragraph now reads as follows: (line 185)

- “In summary, previous research has established that type Ia afferents encode muscle stretch velocity, while type II afferent signalling is more directly associated with static muscle length, although their function overlap to some extent; both position and velocity related signals can be extracted from both afferent types. The behavioural responses to muscle vibration demonstrate that both perceived limb velocity and position are strongly affected; while biased type Ia afferent activity is thought to cause the bulk of the perceptual effects, both type II and Golgi tendons afferents may also contribute to some extent. As such, it remains unknown to what extent proprioceptive inference is cross-modal, such that velocity signals contribute to the perceived position during movement through an integration-like function, thereby providing a signal regarding change in position. This kind of cross-modal sensory inference is consistent with predictive frameworks of sensorimotor control, such as Active Inference and Optimal Feedback Control. While these frameworks differ in their formulation, both assume that position and velocity signals derive from a shared underlying state and must remain congruent, e.g., through smoothness priors placed on higher-order time derivatives (Friston *et al.*, 2010). For example, if the arm is perceived to be stationary (zero velocity), then its position should not be changing; conversely, a change in position implies a corresponding velocity. This internal consistency is a basic requirement of any dynamic state estimator, and is reflected in these frameworks (Wolpert *et al.*, 1995; Todorov & Jordan, 2002; Friston *et al.*, 2010). Despite these theoretical predictions, the extent to which proprioceptive signals are integrated in such a cross-modal fashion remains empirically unexplored.”

- The methods make use of double vibration, however, there is no mention of the current knowledge of the effects of agonist and antagonist vibration together in the introduction See for example Bellan *et al.*, 2016; Bock *et al.*, 2007; Bock & Thomas, 2011; Eschelmuller *et al.*, 2025; Fuentes *et al.*, 2012; Gilhodes *et al.*, 1986; Longo *et al.*, 2009.

- These are indeed very relevant for our use of dual vibration, and we have now incorporated some of these throughout the introduction to introduce the current knowledge regarding dual vibration: (line 113)

- “Studies that have specifically investigated the effects of concurrent vibration of a flexor-extensor muscle pair in non-moving conditions have shown varied effects. Gilhodes *et al.* (1986) found that concurrent vibration tends to cancel each other out when the vibration frequency is kept identical, with effects appearing when the frequency of one motor is decreased, with the faster vibration motor then beginning to produce an effect, although it is smaller than if this vibration motor were used alone. Fuentes *et al.* (2012) and Chancel and Ehrsson (2023) have argued that concurrent vibration may instead degrade the available proprioceptive information, such that variable error increases, and less weight is placed on proprioceptive feedback during

sensory integration, and, as an example, leading to increased belief in the rubber hand illusion.”

- Line (133)

- “As in the passive condition, concurrent agonist-antagonist vibration has been found to have variable effects. Eschelmuller et al. (2025) demonstrated that concurrent vibration produces an effect approximately equal to the sum of their individual effects, with no increase in variable error, congruent with the notion of limb motion being inferred from the combined signalling from all muscles acting on the relevant joint. Bock et al. (2007), on the other hand, found increased errors in angle matching and force production during concurrent vibration, and argued for a general degradation of proprioceptive feedback. However, as they only reported absolute errors, it is not clear to what extent these errors in angle matching and force production arise from decreased mean accuracy in a particular direction, or are due to increases in variable errors, making their results more difficult to place in context. Further, for the angle-matching task, it is not specified how, or whether, muscle thixotropic effects were accounted for during the passive movement of the left arm (which the right arm was supposed to match), leaving open the possibility of bias in the perceived position (Proske & Gandevia, 2012)

Methods:

- What was the monetary value that the participants were compensated?

- The hourly compensation rate has been added.” (line 221)

- “[...] participants] were compensated monetarily for their time at a rate of 160 DKK per hour.”

Task 1:

- What speed were the participants asked to move at? Were they trained at this speed during the practice?

- We have reworded the description of the experiment to highlight that they were not required to move at any particular movement speed, but that they should simply rate the relative movement speed of the seen virtual arm. (line 262)

- “Participants simply needed to move the virtual arm past the target location to end each trial, and were not instructed to move at any particular speed. They were informed that the virtual arm was manipulated to always be slightly slower or faster than the actual speed of their arm (they were not informed of the 'True speed' control condition), and they were instructed to judge whether the movement speed of the virtual arm was 'Faster' or 'Slower' than their actual arm after each trial. “

Task 2: Please include more detail of the task instruction for the participant.

- What is meant by indicate the location of the target by moving their arm and pressing a thumb button.

- Were they aligning their thumb, index finger etc.?

- Were they allowed to make corrections?

- How fast were they told to move?

- We have further specified the task setup and instructions, both in the 'General experimental procedures' and for Task 2 specifically, so as to include these details. (line 277)

- "Participants were instructed to point to the location of the target by moving their arm and aligning their thumb to it before pressing the thumb button on the VR controller. They were not instructed to move at any particular speed, and were allowed to make corrections until the thumb button was pressed, which ended the trial."

Task 3: Please also include more details

- How fast were they told to move?

- We added a paragraph to elaborate on this point. (line 292)

- "During Task 3, participants were again instructed to indicate the target position as precisely as possible by aligning their thumb to the perceived position of the target, which was always presented 80 ° from the start location. Participants were not instructed to move at any particular speed. "

- Please explain/justify why you chose to make the comparisons you have focused on. Why 0-20 and 20-80 degrees.

- We expanded on this point in the methods section: (line 301)

- "We chose to target vibration start/stop at 20 ° into the movement, to trigger the vibration motor to turn on or switch off early enough that any effects on movement velocity would have time to unfold well before the participant ended the trial."

Rotating armrest:

- More detail is needed on the collection of elbow angle/hand position. Was it only the VR controller? What is the sampling frequency of this controller? Spatial resolution?

- We added additional details here, including a new section describing how we have characterised the reliability of the tracking solution.

- (line 325)

- “The controller mounted on the rotating armrest was tracked via the native inside-out tracking of the Meta Quest 3 headset, with the position of the controller reported at 90 Hz. This tracking data was used when visual feedback of the controller location was used in the experiment (Task 1), as well as for our data analysis. As we are not aware of any specific investigations of the tracking performance and characteristics of the Meta Quest 3 controllers, we evaluated it as follows.

In order to track the changes in elbow joint angle, the axis of rotation is required, but it is not tracked directly. We instead estimated it by sampling multiple points along the circle periphery. During each trial, the centre of rotation was re-estimated from 30 equally distributed tracked points along the circle periphery from the previous trial, captured as the participant rotated the armrest from the starting position to the target position. This point was re-estimated in each trial to guard against any potential tracking drift, or if the table itself was nudged slightly during a trial, moving the actual axis of rotation. This method of estimating the axis of rotation provided a fairly stable estimate of the axis in the tracked space, with a trial-to-trial standard deviation of 1.2 mm, 0.63 mm and 0.85 mm in the x (lateral), y (vertical), and z (sagittal) directions, respectively. Some of the stability comes from averaging the circle centre calculated from across 30 points along the perimeter. We further checked the mean within-participant trial-to-trial standard deviation of the estimated distance from this circle centre to the controller, with the tracked controller location taken from a single sample in each trial, in order to give some measure of reliability when not averaging across tracking points; as the controller was hard-mounted to the armrest, this distance is constant for a given participant. This provided a mean within-participant trial-to-trial standard deviation of 0.29 mm distance. The group mean distance was estimated at 36.1 cm (range 31.6 – 42.8 cm), corresponding to the elbow to mid-hand distance. While this is not a measure of the true tracking error (i.e., there could be consistent biases in specific directions), it nevertheless highlights that the tracking solution is quite repeatable and stable in performance.”

Vibration motors:

- If you wanted to avoid changes in pressure, why wasn't the triceps vibrator also secured onto the muscle belly?

- We have now provided a more thorough description of our approach: (line 369)

- “One vibration motor was mounted over the distal triceps tendon, just proximal to the elbow, while the other was mounted centrally on the belly of the biceps muscle. While vibration experiments most often target the muscle tendons rather than the muscle belly directly, previous research has indicated no difference in the required vibration amplitude to elicit an illusory movement between the distal biceps tendon and the biceps muscle belly (Ferrari *et al.*, 2019).

The mounting of both motors was chosen to minimise pressure changes during elbow flexion/extension; with the motors mounted to the arm via tape, the muscles are able to slide under them, substantially altering the circumference of the arm at the mounting point, if the muscle is sometimes under the motor, and sometimes not.

We found the best compromise here was to mount the biceps motor centrally on the muscle belly, such that the biceps muscle would never shorten enough to no longer be under the motor. The triceps motor was instead mounted over the distal tendon, such that the triceps would never lengthen enough to slide under the motor. We were unable to find a pair of mounting spots where both motors would consistently target either the muscle belly or tendon throughout the whole flexion-extension range. Pilot testing indicated that the chosen mounting spots nevertheless produced the expected proprioceptive illusions in both flexion and extension movements.”

- What was the amplitude of the vibration?

- We now describe the following regarding the amplitude: (line 356)

- “The vibration amplitude was not measured during the experiment and must be expected to continually change somewhat during movement due to changes in mounting pressure, as well as from participant to participant; in positions where the tape holds the motor more firmly against the skin, the effectively vibrated mass increases, leading to decreased vibration amplitude of the motor itself. This is a distinct disadvantage compared to larger externally hard-mounted vibration motors; instead, they allow for greater and more comfortable movement.“

- The make/model is present but also explain what type of vibrator was used. For example, was it a linear motor, eccentrically mounted DC motor etc. Each will have different characteristics.

- We have added additional details regarding the motor type: (line 351)

- “Two small vibration motors (Vp216 VIBRO transducer, Acouve Laboratory Inc., Tokyo, Japan) were mounted on the right arm, with inelastic micropore tape. This is a coin-shaped voice-coil type vibration motor, with a sprung oscillating mass, driven by an amplifier similarly to loudspeakers. They have a weight of 49 g, a diameter of 43 mm, and a height of 15 mm. When mounted flat on the arm, the vibration direction is in the normal direction from the skin surface, providing a pushing force directly downwards during vibration.“

Data processing and statistical modeling

- Overall, the modeling needs to be more clearly justified and explained for the reader. Why were Bayesian models chosen, and how do they directly inform the study's hypotheses? Why was a Bayesian as opposed to a frequentist approach used? As written, a reader who is not already familiar with the brms ecosystem will struggle to follow the modeling section. The

main manuscript text should describe the models in clear, well-defined, and broadly accepted statistical terms, not in package-specific code syntax. For example, instead of presenting code snippets, the authors could provide the underlying model equations (e.g., linking predictors and outcomes in regression form) or a schematic/diagram that illustrates the flow of the model. Since the exact code is published alongside the manuscript, the main text should focus on conceptual clarity: what question each model addresses, why it is structured that way, and how it connects to the theoretical framework.

- We now write the following regarding the choice of Bayesian methods: (line 393)

“We chose to employ Bayesian regression modelling to analyse our data, as we believe it provides a more straightforward interpretation of parameter estimates, compared to analogous frequentist models. Our models are specified so as to provide directly interpretable parameter estimates. In addition, the Bayesian framework naturally supports partial pooling, which appropriately accounts for the nested structure of trials within participants. We present the regression models here in a minimal form, focusing on readability and omitting the specification of the nested structure. For all models, we included individual-level intercepts and slopes for all the same group-level parameters that are presented here. Full model specifications and priors are available in the code repository.”

- Overall, the models themselves seem technically correct to my knowledge, but this section needs to be significantly reworked with the goal of allowing the reader to understand the rationale and purposes of each model.

- We have significantly reworked this section to provide a clearer description of our goals with each specific model, as well as rewritten the models to more standard formulations, with the goal of making them more easily accessible to a broader audience. As the entire section has almost been completely rewritten, we will here include just some examples.

- As an example, for the task 1 we have now written the following, with a similar style change made for Task 2 and 3 (see the revised manuscript for these): (line 419)

“In Task 1, we investigated the effect of muscle vibration on the perceived and actual movement speed. Both models for these two analyses were implemented in brms as hierarchical (multilevel) models with the random effects defined at the participant level.

Our goal with the first model (Eq. 1) was to investigate whether the vibration motors influence perceived movement speed; we model the distribution of ratings of whether the virtual arm was rated to be ‘Faster’ or ‘Slower’ than the participant’s true arm. Specifically, our hypothesis here was that muscle vibration may change the perception of the movement speed of the executed elbow extension movement; for example, antagonist vibration was expected to increase the perceived velocity of biceps muscle stretch. We probe this perception of movement speed by having the participant report the relative movement speed of the virtual arm. If the antagonist muscle vibration increases the proprioceptively signalled movement speed, we should expect to see an increase in the probability of rating the virtual arm as slower. By further including the actual changes in the visual speed gain of the virtual arm, we can confirm whether the participants answer

as expected when an actual speed gain is applied to the virtual arm, while also establishing a scale of known movement speed changes to judge parameter estimates against.

$$\begin{aligned} rating_i &\sim \text{Bernoulli}(\Phi(z_i)) \\ z_i &= \beta_0 + \beta_1 vg_i + \beta_2 agovib_i + \beta_3 antavib_i \end{aligned}$$

Eq. 1

We modelled this as a Bernoulli model with a probit link, with the visual speed gain (vg), agonist vibration ($agovib$) and antagonist vibration ($antavib$) as predictors. This model assumes that the answer distribution for each condition can be approximated by a latent, z-normalised Gaussian distribution (z_i^* below), with a static threshold at 0 defining whether 'Faster' or 'Slower' is selected:

$$\begin{aligned} z_i^* &= \beta_0 + \beta_1 vg_i + \beta_2 agovib_i + \beta_3 antavib_i + \epsilon_i \\ \epsilon_i &\sim \mathcal{N}(0,1) \\ rating_i &= \begin{cases} \text{'Faster'}, & \text{if } z_i^* > 0 \\ \text{'Slower'}, & \text{if } z_i^* \leq 0 \end{cases} \end{aligned}$$

This latent distribution can be thought to represent the result of an internal comparison between the proprioceptively perceived speed of the actual arm and the visually observed speed of the virtual arm. On any particular trial, both the proprioceptively perceived speed or visually observed speed may be either over- or underestimated due to sensory noise, but across trials, we should expect a subject with no bias from either proprioception or vision to report the virtual arm as 'Faster' on approximately 50 % of trials, when no vibration or visual speed gain is applied; this corresponds to an expected estimate of the intercept (β_0) of around 0 (z_i^* would then be equally distributed around the threshold at 0). The parameter estimate of e.g., β_3 then describes how much of the latent Gaussian is shifted relative to β_0 when antagonist vibration is applied. Note that visual speed gain is included as a linear predictor on the latent scale, such that +12% visual speed gain is expected to have twice the effect of +6%.

This type of model is closely related to classical Signal Detection Theory models (Stanislaw, 1999), with the conceptual difference that the signal here (relative speed of the virtual arm) may be either higher or lower than 0.

We additionally modelled any changes in trial-wise mean movement speed (mms , in °/s) in Eq. 2. For each participant, mms was centred relative to the mean in the no-vibration / normal-visual-feedback condition, because we are primarily interested in changes in speed due to muscle vibration rather than baseline differences between participants. Our goal with this model was specifically to investigate whether muscle vibration causes any changes in actual movement, which complements our above model, which examined the effects on the perceived movement speed.

Target distance (td) is included here as a predictor because it strongly predicts variability in movement speed; including td thus reduces unexplained variance and thereby improves the precision of estimates of the vibration and visual-gain effects.

$$\begin{aligned} mms_i &\sim \mathcal{N}(z_i, \sigma) \\ z_i &= \beta_0 + \beta_1 vg_i + \beta_2 agovib_i + \beta_3 antavib_i + \beta_4 td_i \end{aligned}$$

Eq. 2

360 trials were completed per participant, except for one, for whom the last 194 trials could not be completed due to technical issues. This leaves a total of 5926 included trials for both models.”

Results:

- Overall, the results were not fully presented and explained. Many details have been omitted, and some analyses are never mentioned even though they are present in some of the figures.

- We have now substantially reworked the presentation of the results; together with the more thorough presentation of the statistical models in the Methods section, we hope that the results section is now more easily followed.

- We will provide some specific examples in the responses to the following comments.

- Task 1 results. There is no discussion or visualization of the effects of the double vibration results here. Additionally, it is unclear why only the visual speed gain of 6% is visualized in 2B and 2C. In figure 2B, the reference is no vibration. Presumably this was from the visual speed gain of 100%. Similarly, were the agonist and antagonist vibration errors also calculated based on the 100% relative movement speed. Potentially this information could be inferred from the modelling section above, but as mentioned earlier this section is not very clear and it should be explicitly stated here.

- We originally omitted the visualisation of the double vibration condition to emphasise the observed effects of each vibration motor. Upon review, we agree that this was an oversimplification, and we have now amended the plot (**Figure 2: Task 1 results**) to show the estimated effect, and added additional description of how these results may be interpreted. Along with the much-expanded introduction of the used Bernoulli model in the Methods section, we hope that the presentation of the analysis results is now clear and interpretable.

- We now write: (line 623)

“As shown in Figure 2A, we found a 51.6% (95% CI [42.1, 60.6]) probability of rating the virtual arm as 'Faster', in the control condition with no vibration and no visual speed gain, indicating that the average estimation of relative movement speed is correct. Varying the visual speed gain from -12% to +12%, shows the expected shift in the probability of reporting the virtual arm as 'Faster'; when the visual speed gain is increased, the probability of reporting it as such increases, and vice versa when it is decreased. This effect was estimated with a linear coefficient of 0.41 (95% CI [0.31, 0.52]) for a visual gain of +6% (purple line interval, Figure 2B). This estimate corresponds to the 'slope' of the black line in Figure 2A; this variable is modelled linearly on the probit latent scale, producing the observed S-shaped curve in the visual-speed-gain-to-probability plot. This leads to a group mean probability of correctly rating the virtual arm as faster of 67.2% (95% CI [56.3, 77.2]) for the +6%

condition and 80.2% (95% CI [68.8, 89.2]) for the +12% condition, with similar changes in the opposite direction for the negative speed gains.

Having thus validated that the participants can successfully compare their own arm movement speed to that of the virtual arm, we can interpret the effects of applying muscle vibration during this task. Here, we found a decrease in the probability of reporting the virtual arm as 'Faster' when antagonist vibration is applied, with a coefficient of -0.29 (95% CI [-0.47, -0.1]), which corresponds to a decrease to 40.3% (95% CI [31.2, 49.6]) probability of reporting the virtual arm as faster when no visual speed gain is applied. This may then be interpreted as antagonist muscle vibration causing participants to overestimate their movement speed, as they here become less likely to report the virtual arm as 'Faster'. For further visualisation of the underlying rating data, see Figure A1 in the appendix.

In addition to biasing the perceived movement speed, antagonist vibration also causes adjustments of the actual movement speed; as the participants believe they are moving faster, they also decrease their mean movement speed by 1.94 °/s (95% CI [0.55, 3.44]).

For agonist vibration, the results are mixed. While the actual movement speed increased by 1.41 °/s (95% CI [0.13, 2.88]), the 95% credible interval describing the change in probability of reporting the perceived movement speed overlapped 0, indicating insufficient evidence to claim such an effect. Concurrent vibration of both motors tended to cancel each other out, such that the perceived relative speed and actual movement speed remained unchanged compared to the no vibration condition.

“

- Task 2 results. This section is difficult to interpret as many details are missing from the task instruction and parameters in the methods. In the first line, figure 6 is reference, however this seems incorrect. As with task 1, the result of the double vibration is not sufficiently reported.

- The ref to figure 6 was indeed in error, and have now been corrected. We hope that the additional details provided in the Methods section, as well as the expanded description of the results here in the results section make the analysis outcomes easier to follow.

- As an example, we now write the following regarding the effects of vibration motors on endpoint errors, with similar adjustments made to the reporting of changes in movement speed and movement time: (line 674)

“In Task 2, we investigated how muscle vibration affects movement time, movement speed, and endpoint errors (Figure 3). While we focus on the changes in performance caused by muscle vibration in the following section, a summary of the baseline performance in the 'No vibration' trials is provided in Table A1 for reference in the appendix.

We found that antagonist vibration caused a consistent bias towards undershooting the target, both in the flexion (-2.76 ° (95% CI [-3.8, -1.64])) and extension directions (-1.32 ° (95% CI [-2.07, -0.48])). Agonist vibration, in contrast, caused participants to

overshoot the target in the flexion direction (1.45° (95% CI [0.51, 2.39])), but not in the extension direction. The estimated interaction effect of agonist and antagonist vibration together overlapped zero, indicating that their combined effect is not different from the sum of each effect alone; this produced a resulting undershooting effect of double vibration during flexion, but not extension movements. Furthermore, we found no indication of increased standard deviation of endpoint errors, suggesting that neither agonist, antagonist, nor both of them together have any effects on movement precision (Figure 3D). “

- Task 1 and 2 integrative results. Here the authors again state that position estimates are derived by integrating sensed movement speed. However, if this were the case, the correlations in figure 4 would be expected to be much stronger. If the end point errors are being driven by changes in movements speed, we should see robust correlations between the two measures. Tendon vibration likely drives changes in both perceived position of the hand and perceived velocity. The authors argument that both are derived by velocity is not grounded in previous work or their current data. The authors need to outline this idea in more detail (as mentioned above in the introduction), but also here based on the data that they have, as currently it is not clear.

- We have now substantially reworked this section to provide a more nuanced description and interpretation of the analysis here, with the goal of highlighting that we do not suggest that position is solely inferred from velocity sources, but rather that it may be a contributing source of information for state estimation. Furthermore, the goal of this analysis of the baseline control trials is to emphasise that it appears that the multiple different effects of muscle vibration are interrelated, and are apparent even in the absence of any external biasing effects. We hope that this is now clear in the text.

As an example, we now start the section with the following: (line 741)

“One way to jointly interpret the above findings from Tasks 1 and 2 is to assume that the CNS estimates limb state in a cross-modal fashion, with position- and velocity-based proprioceptive signals augmenting each other and contributing to a shared state estimate, as in predictive frameworks of sensorimotor control. We further explore this notion by considering the subset of baseline control trials from Tasks 1 and 2 with no vibration and no visual speed gain, as we should expect to see this same effect reflected here, and not only perturbed trials.”

- And we now end the section: (line 798)

- “From Figure 5B, it is clear that there is substantial uncorrelated variability in movement speed and endpoint errors; we do not here suggest that all inferred position information originates from velocity feedback sources, but rather that inferred velocity does appear to contribute meaningfully to the perception of position. “

- See the revised manuscript for the full reworked section.

Discussion

- Due to issues with data presentation and reporting of task parameters, I will only provide a high-level review of the discussion as without the extra details it is difficult to get a whole picture of the experiments. As with the discussion, the effects of dual vibration are essentially left out completely.

- We have substantially reworked the discussion, and the same paragraph structure no longer fully applies. But we will in the give examples that highlight where we address each of the comments.

- 5.1 paragraph 3. The authors need to expand their viewpoint that vibration only biases the "velocity sensors". They bias both Ia and group II afferents, both of which can signal velocity and position. The relative amount of signal from each depends on the gamma motoneuron bias present.

- We now write the following: (line 840)

“In Task 1, our results show that antagonist muscle vibration causes participants to overestimate their movement speed while simultaneously decreasing their actual movement speed, whereas agonist muscle vibration increases movement speed. While the perception of movement speed is only reported after the trial, it seems likely that the changes in movement speed arise as a corrective response to biased velocity feedback. This effect seems likely to be caused by the action of muscle vibration on the muscle stretch-velocity sensing capabilities of the muscle spindle afferents. While both type Ia and type II afferents include velocity components in their signalling (Roll & Vedel, 1982), the primary perceptual effects of muscle vibration on the perception of movement is believed to be caused by type Ia afferent activity (Proske & Gandevia, 2012; Banks et al., 2021).

[...]

It might further be considered if some of the changes in movement speed could be caused by a direct effect of muscle vibration biasing the sensed limb position; e.g. antagonist vibration could bias the sensed limb position through the effects on both types of afferents, such that the arm is sensed to be further along the movement than is the case, essentially adding a static offset to the perceived position. Thus, sensing a position closer to the target than anticipated might similarly cause a reactive reduction in movement speed. However, it is less clear how this effect would fit together with our findings of the associated changes in the perception of movement speed.”

- Overall section 5.1 doesn't provide a very detailed discussion of the data they collected and how this links back to their original hypotheses and the models they mention (OFC and AI).

- We now expand on this point: (line 872)

“By further considering our analysis of the subset of trials with no vibration and normal visual feedback, we observe that variability in movement speed is generally predictive of the reported perception of movement speed. Specifically, trials performed with higher movement speeds are associated with underestimating the movement speed of the arm, and vice versa. This supports the notion that the effects of muscle vibration on both perceived and actual movement speed are not two concurrent but independent effects, both caused by muscle vibration. Rather, it appears that these two effects are indeed interdependent; it seems likely that as sensed movement speed becomes biased, a corrective speedup or slowdown is performed, perhaps to match an intended movement speed profile. Movement speed profiles of reaching movements are typically bell-shaped and smooth, a feature that has been well-accounted for by models based on signal-dependent noise and Optimal Feedback Control (Todorov & Jordan, 1998; Harris & Wolpert, 1998), as well as through models that minimise variational free energy (Friston et al., 2010). From either of these perspectives, any error in the perception of movement speed, whether arising spontaneously or through muscle vibration, would be expected to trigger the observed compensatory motor adjustments. These may align ongoing movement with either a planned optimal trajectory or with predictions produced by a generative model, depending on the assumed control framework.”

- Section 5.2 requires the data to demonstrate that changes in movement speed cause a change in end point position. In my opinion the authors have not clearly shown this cause-and-effect relationship. It could be that vibration biased both perceived position and movement, and both are used to determine end point. The correlations that have been presented in figures 4 and 5 are not convincing enough to make these statements. The discussion of the paradoxical effects on movement time is not clear, and the authors should provide a more detailed explanation on how these fits with their model of the task and results.

- We have reworked this section to provide a more nuanced discussion of the results. In particular we here aim to highlight that we do not intend to conclude that all position estimation is a downstream effect of velocity-based feedback, but rather that velocity feedback may be used to augment positional inference, as a signal regarding change in position over the course of a movement.

- We here highlight the difficulty in separating the possible effects of both position and velocity based feedback: (line 910)

- “It should, however, be noted that both type Ia and type II muscle spindle afferents also provide position-based signals, which are also likely to be affected to some degree as well by muscle vibration. As such, it is difficult to ascertain to what extent the errors in perceived position revealed through the endpoint errors are a downstream

effect of the integration of velocity signalling, and to what extent they are more direct effects of biased position sensors.”

- Regarding the observed effects on movement time, we now write: (line 947)

“During active control of movement, the CNS appears to rely on inferred movement velocity to trigger movement sequences at the correct timing to account for sensory and motor delays inherent to nervous conduction and processing (Cordo, 1988; Cordo et al., 1994). For example, opening the hand at the correct arm angle during an arm extension movement, as is required for accurate throwing, requires accounting for both efferent and afferent delays; if the command to open the hand is sent only once the afferent feedback from the arm communicates that the target angle has been reached, then the hand would be opened too late, by an amount corresponding, at least, to the sum of the afferent and efferent delays. Such observations are well in line with predictive frameworks of sensorimotor control, where internal generative models are thought to bridge the temporal gap between delayed sensory feedback and efferent motor output (Todorov & Jordan, 2002; Friston, 2011). These timing and coordination effects pose an additional challenge for interpreting how muscle vibration affects the currently inferred position and velocity, and how the inferred position is expected to change in the immediate future. For example, it might well be imagined that even if an induced velocity offset does not affect the currently perceived position, it might still affect a forward prediction about the future position of the hand, prompting a different timing of pushing the button to indicate the position of the target. However, our analysis reveals small but consistent changes in movement time that work in the opposite direction. For example, if antagonist vibration affected only the forward-predicted timing of reaching the target, we should expect to see decreased movement time; however, to the extent that we do see any effects on movement time, they are in the opposite as to what would be expected, if the above describing change in timing was a primary contributor to the changes in endpoint error that we see. Generally, in a simplified sense, undershooting the target might have been associated with either moving slower for an unchanged movement time or moving for a shorter time at a similar speed. Instead, we tend to see increased movement time in response to antagonist vibration, effectively increasing the required decrease in movement speed to achieve the same shift in endpoint error towards undershooting the target, and vice versa for agonist vibration. As it is, we are not able to provide any explanation for the observed changes in movement time, and further research is warranted to explore how these effects fit in the overall pattern of effects.”

Referee #3:

Proprioceptive integration in motor control

Authors: Erik Mortensen and Mark Christensen

I read this paper with great interest. I believe it presents a coherent set of well-designed experiments on the role of Ia afferent feedback on control and estimation, using muscle vibration. Also, the statistical methodology is powerful, using MCMC-techniques to unveil group level effects. This methodology is very well suited to analyse the highly variable responses to muscle vibration. The description of these methods well done and provides enough detail to reproduce. So, overall a great paper that I would recommend publishing.

- We greatly appreciate the positive feedback. We have responded to each of the specific comments below.

Here some minor remarks:

[General experimental procedures] What is the precision of the alignment between the actual arm and the virtual arm and how was this established?

- We have added a paragraph in the Methods section which describes how we have characterised the reliability of the tracking solution to display the virtual arm: (line 330)
- “In order to track the changes in elbow joint angle, the axis of rotation is required, but it is not tracked directly. We instead estimated it by sampling multiple points along the circle periphery. During each trial, the centre of rotation was re-estimated from 30 equally distributed tracked points along the circle periphery from the previous trial, captured as the participant rotated the armrest from the starting position to the target position. This point was re-estimated in each trial to guard against any potential tracking drift, or if the table itself was nudged slightly during a task, moving the actual axis of rotation. This method of estimating the axis of rotation provided a fairly stable estimate of the axis in the tracked space, with a trial-to-trial standard deviation of 1.2 mm, 0.63 mm and 0.85 mm in the x (lateral), y (vertical), and z (sagittal) directions, respectively. Some of the stability comes from averaging the circle centre calculated from across 30 points along the perimeter. We further checked the mean

within-participant trial-to-trial standard deviation of the estimated distance from this circle centre to the controller, with the tracked controller location taken from a single sample in each trial, in order to give some measure of reliability when not averaging across tracking points; as the controller was hard-mounted to the armrest, this distance is constant for a given participant. This provided a mean within-participant trial-to-trial standard deviation of 0.29 mm distance. The group mean distance was estimated at 36.1 cm (range 31.6 – 42.8 cm), corresponding to the elbow to mid-hand distance. While this is not a measure of the true tracking error (i.e., there could be consistent biases in specific directions), it nevertheless highlights that the tracking solution is quite repeatable and stable in performance.”

Task 1: Smart way of investigating the influence of vibration on movement speed.

Task 2: Manipulation to investigate vibration effects on actual movement speed changes.

Task 3: Time/position dependent changes in movement speed.

I like the sophisticated statistics with partial pooling in BRMS and the centered approach.

Figure 2: I would prefer visual speed gain as -12%, -6%, 6% and 12%, so the change, as this is how you report in the main text.

- We have revised the figure axis text to match this wording used in the main text.

Part 4.2 - do you really intend to refer to Figure 6 here? Then maybe move the figure, you now make a very far fast forward.

- The reference to Fig 6 here was indeed in error, and this has now been corrected.

Part 5.3 - You compare the behavioural onset latency wrt vibration onset, to EMG based measures. However, Keyser et al. (2019) have also reported behavioural latencies, so good to mention that you find them in the same range.

- The reference article by Keyser et al. was indeed very relevant to our findings, and we have incorporated it in our discussion of our results: (line 995)
- “In Task 3, our results show how movement speed is affected immediately following turning on or off the antagonist vibration motor. The expected decrease in movement speed in response to antagonist vibration had an onset latency of approximately 50 ms. This finding is well in line with the findings by Keyser et al. (2019), who demonstrated a latency of changed force-production at 62 ms after onset of biceps vibration, and similarly matches EMG-based research, which has identified changes in muscle activity at latencies of 40-60 ms (Capaday & Cooke, 1983; Cody et al., 1990). Our observations further expand on this work, demonstrating that the previously observed latency of muscle force and EMG response matches well with actual changes in movement speed. “

"From a signal-dependent noise perspective ..." - this sentence doesn't make sense. The mentioned framework doesn't include corrections. OFC is a signal-dependent noise feedback control framework and that explicitly does not restore the original trajectory. If you want to make a case for trajectory control in OFC, you should have a look at: Cluff, T., & Scott, S. H. (2015). Apparent and actual trajectory control depend on the behavioral context in upper limb motor tasks. *Journal of Neuroscience*, 35(36), 12465-12476.

- Thank you for the feedback and suggestions. Upon review, we have removed this sentence.

Dear Dr Mortensen,

Re: JP-RP-2025-289835R1 "Proprioceptive integration in motor control" by Erik Skjoldan Mortensen and Mark Schram Christensen

Thank you for submitting your revised Research Article to The Journal of Physiology. It has been assessed by the original Reviewing Editor and Referees and has been well received. Some final revisions have been requested.

REVISION CHECKLIST:

We look forward to receiving your revised submission.

Yours sincerely,

Vaughan Macefield
Senior Editor
The Journal of Physiology

REQUIRED ITEMS

- Papers must comply with the Statistics Policy: https://jp.msubmit.net/cgi-bin/main.plex?form_type=display_requirements#statistics.

In summary:

- If n {less than or equal to} 30, all data points must be plotted in the figure in a way that reveals their range and distribution. A bar graph with data points overlaid, a box and whisker plot or a violin plot (preferably with data points included) are acceptable formats.
- If $n > 30$, then the entire raw dataset must be made available either as supporting information, or hosted on a not-for-profit repository, e.g. FigShare, with access details provided in the manuscript.
- 'n' clearly defined (e.g. x cells from y slices in z animals) in the Methods. Authors should be mindful of pseudoreplication.
- All relevant 'n' values must be clearly stated in the main text, figures and tables.
- The most appropriate summary statistic (e.g. mean or median and standard deviation) must be used. Standard Error of the Mean (SEM) alone is not permitted.
- Exact p values must be stated. Authors must not use 'greater than' or 'less than'. Exact p values must be stated to three significant figures even when 'no statistical significance' is claimed.

EDITOR COMMENTS

Reviewing Editor:

The authors have made substantial revisions to their submission which improve the clarity of their work and make it easier to appreciate their findings and what they mean.

Senior Editor:

Thank you for submitting your revised manuscript to The Journal of Physiology. The Reviewing Editor and I are satisfied that you have addressed the concerns raised by the external reviewers. However, there are some small amendments I need you to make before we can accept your paper.

In the Introduction, when describing muscle spindle afferents, please change "...where they innervate intrafusal muscle fibres (Proske & Gandevia, 2012)" to "...where they innervate the non-contractile central region of the intrafusal muscle fibres (Proske & Gandevia, 2012)"

Please change "It should further be noted that the Golgi tendon organ, in addition to both types of muscle afferents, is also

affected by vibration (Roll et al., 1989)" to "It should further be noted that the Golgi tendon organ, in addition to both types of muscle afferents, is also affected by vibration when the muscle is contracting (Roll et al., 1989)"

"These behavioural findings align with microneurographic studies of type Ia afferent activity. At rest, type Ia afferents exhibit minimal firing rates, with firing rate increasing proportionally to the velocity of muscle lengthening (Matthews, 1963; Roll & Vedel, 1982; Roll et al., 1989)." Please note that Matthews studied muscle spindles in the cat; microneurographic studies were only conducted in humans. I would change "microneurographic studies of type Ia afferent activity" to "recordings of type Ia afferent activity in the cat and humans"

REFEREE COMMENTS

Referee #3:

I stick with my earlier statement, that the contribution is valid and thorough but not surprising.

END OF COMMENTS

Response to editors:

We greatly appreciate the detailed reading of the manuscript and the provided feedback. We have incorporated all three suggested revisions.

Additionally, figure texts have been updated to also include relevant 'n' information per the statistics policy (previously only present in Methods section).

Reviewing Editor:

The authors have made substantial revisions to their submission which improve the clarity of their work and make it easier to appreciate their findings and what they mean.

Senior Editor:

Thank you for submitting your revised manuscript to The Journal of Physiology. The Reviewing Editor and I are satisfied that you have addressed the concerns raised by the external reviewers. However, there are some small amendments I need you to make before we can accept your paper.

In the Introduction, when describing muscle spindle afferents, please change "...where they innervate intrafusal muscle fibres (Proske & Gandevia, 2012)" to "...where they innervate the non-contractile central region of the intrafusal muscle fibres (Proske & Gandevia, 2012)"

- The suggested revision has been incorporated.

Please change "It should further be noted that the Golgi tendon organ, in addition to both types of muscle afferents, is also affected by vibration (Roll et al., 1989)" to "It should further be noted that the Golgi tendon organ, in addition to both types of muscle afferents, is also affected by vibration when the muscle is contracting (Roll et al., 1989)"

- The suggested revision has been incorporated.

"These behavioural findings align with microneurographic studies of type Ia afferent activity. At rest, type Ia afferents exhibit minimal firing rates, with firing rate increasing proportionally to the velocity of muscle lengthening (Matthews, 1963; Roll & Vedel, 1982; Roll et al., 1989)." Please note that Matthews studied muscle spindles in the cat; microneurographic studies were only conducted in humans. I would change "microneurographic studies of type Ia afferent activity" to "recordings of type Ia afferent activity in the cat and humans"

- The suggested revision has been incorporated.

Dear Mr Mortensen,

Re: JP-RP-2026-289835R2 "Proprioceptive integration in motor control" by Erik Skjoldan Mortensen and Mark Schram Christensen

We are pleased to tell you that your paper has been accepted for publication in The Journal of Physiology.

IMPORTANT REQUEST

We have a note to check with you that you have made your data open so that you are eligible for a Wiley Open Science Badge on publication. Please confirm by email to jp@physoc.org that you have made your data open. Once we have confirmation from you, we will send your accepted article over to production. Thank you!

Yours sincerely,

Vaughan Macefield
Senior Editor
The Journal of Physiology

IMPORTANT POINTS TO NOTE FOLLOWING ACCEPTANCE OF YOUR PAPER:

- **IMPORTANT NOTICE ABOUT OPEN ACCESS:** To assist authors whose funding agencies mandate immediate public access to published research findings, The Journal of Physiology allows authors to pay an Open Access (OA) fee to have their papers made freely available immediately on publication.

- You can help your research get the attention it deserves! Check out Wiley's free Promotion Guide for best-practice recommendations for promoting your work at: www.wileyauthors.com/eoo/guide. You can learn more about Wiley Editing Services which offers professional video, design, and writing services to create shareable video abstracts, infographics, conference posters, lay summaries, and research news stories for your research at: www.wileyauthors.com/eoo/promotion.

- If you would like to receive our 'Research Roundup', a monthly newsletter highlighting the cutting-edge research published in The Physiological Society's family of journals (The Journal of Physiology, Experimental Physiology, Physiological Reports, The Journal of Nutritional Physiology and The Journal of Precision Medicine: Health and Disease), please click this link, fill in your name and email address and select 'Research Roundup':

<https://www.physoc.org/journals-and-media/membernews>

EDITOR COMMENTS

Thank you for incorporating these recommendations. I am pleased to report that your manuscript is now considered acceptable for publication in The Journal of Physiology.